# Electrocatalytic water oxidation with manganese phosphates

Shujiao Yang [1,3], Kaihang Yue [2,3], Xiaohan Liu [1], Sisi Li [1], Haoquan Zheng [1], Ya Yan [2] ✉, Rui Cao [1] & Wei Zhang [1] ✉

As inspired by the $Mn_4CaO_5$ oxygen evolution center in nature, Mn-based electrocatalysts have received overwhelming attention for water oxidation. However, the understanding of the detailed reaction mechanism has been a long-standing problem. Herein, homologous $KMnPO_4$ and $KMnPO_4•H_2O$ with 4-coordinated and 6-coordinated Mn centers, respectively, are prepared. The two catalysts constitute an ideal platform to study the structure-performance correlation. The presence of Mn(III), Mn(IV), and Mn(V) intermediate species are identified during water oxidation. The Mn(V)=O species is demonstrated to be the substance for O−O bond formation. In $KMnPO_4•H_2O$, the Mn coordination structure did not change significantly during water oxidation. In $KMnPO_4$, the Mn coordination structure changed from 4-coordinated $[MnO_4]$ to 5-coordinated $[MnO_5]$ motif, which displays a triangular biconical configuration. The structure flexibility of $[MnO_5]$ is thermodynamically favored in retaining Mn(III)−OH and generating Mn(V)=O. The Mn(V)=O species is at equilibrium with Mn(IV)=O, the concentration of which determines the intrinsic activity of water oxidation. This study provides a clear picture of water oxidation mechanism on Mn-based systems.

Converting solar energy into chemical energy via artificial photosynthesis is an effective way to solve energy and environmental problems[1–5]. Water oxidation is the key reaction in artificial photosynthesis[6–9] and is sluggish because of kinetic and thermodynamic challenges[10,11]. Inspired by the biological $Mn_4CaO_5$ cluster for efficient water oxidation in natural photosystems, considerable efforts have been devoted to developing efficient Mn-based artificial catalysts[12,13]. However, it is still a key and challenging issue to explore the relevant reaction mechanism[14–16].

Heterogeneous electrocatalytic water oxidation is an intricate interface electrochemical reaction[17–20]. The reaction itself involves complex electron/proton transfer and oxygen-oxygen bonding[21–24]. In addition, the chemical valence state of manganese is mutable[25]. All these features make the reaction mechanism studies of Mn-based catalysts extremely difficult[26]. At present, Mn-based material catalysts reported in literature often exhibit various differences in valence state[27], structure[28], interlayer spacing[29], Mn vacancy[30], morphology[28,31], crystallinity[32–34], surface state[25], and so forth. Multiple interdependent parameters can simultaneously affect the catalytic performance. Therefore, it is usually unlikely to obtain the immediate cause of the activity by comparing different Mn-based systems, let alone the specific catalytic mechanisms. Herein, we plan to prepare homologous manganese phosphate catalysts with clear crystal structures, focusing on regulating the coordination configuration of manganese and avoiding introducing other interference factors. Through building an ideal comparative study platform, we propose to explore the structure-activity relationship and to obtain a clear picture of water oxidation mechanism in Mn-based systems.

In the present study, we synthesized 4-coordinated ($KMnPO_4$) and 6-coordinated ($KMnPO_4•H_2O$) manganese phosphates as electrocatalysts for the oxygen evolution reaction (OER). The former is derived from the dehydration of the latter. Thus, the two materials

[1]Key Laboratory of Applied Surface and Colloid Chemistry, Ministry of Education; School of Chemistry and Chemical Engineering, Shaanxi Normal University, Xi'an 710119, China. [2]Shanghai Institute of Ceramics, Chinese Academy of Sciences (SICCAS), Shanghai 200050, China. [3]These authors contributed equally: Shujiao Yang, Kaihang Yue. ✉e-mail: yanya@mail.sic.ac.cn; zw@snnu.edu.cn

have many physical similarities except the core coordination structure of manganese. Electrocatalytic results indicate that the KMnPO₄ shows much better activity than the monohydrate $KMnPO_4 \cdot H_2O$. With ex-situ and in-situ characterizations, Mn(III) and Mn(IV) active intermediates were confirmed in both catalysts during OER. The oxidation events of Mn(II) to Mn(III) and Mn(III) to Mn(IV) are 1H⁺/1e processes. The active species before the O−O bond formation were confirmed to be [MnO₅] and [MnO₆] motifs with high valent Mn(V)=O site. The Mn(V)=O species was at equilibrium with Mn(IV)=O, the concentration of which determined the intrinsic activity of water oxidation. The changes of the coordination environment of the two catalysts during OER were analyzed by in-situ X-ray absorption spectra (XAS) studies. In the 4-coordinated KMnPO₄, the Mn coordination structure changed at an early stage into 5-coordinated [MnO₅] motif, which displays a triangular biconical configuration. The 5-coordination geometric structure remained unaltered but the atomic position underwent changes during water oxidation. The structure flexibility of [MnO₅] is thermodynamically favored in retaining Mn(III)−OH and generating Mn(V)=O, which leads to its higher intrinsic activity for water oxidation.

## Results and discussion

### Physical characterizations of the electrocatalysts

The $KMnPO_4 \cdot H_2O$ (6-coordinated) was prepared by a co-precipitation method (for details see the Methods)[35,36]. The KMnPO₄ (4-coordinated) was prepared by heat treatment of $KMnPO_4 \cdot H_2O$ at 300 °C for 1 h. The surface morphologies of the two catalysts were characterized by scanning electron microscope (SEM), showing similar nanosheet structures (Fig. 1a, b and Supplementary Fig. 1). The transmission electron microscopy (TEM) (Fig. 1c, e), high-resolution TEM (HR-TEM) (Supplementary Fig. 2), and the atomic force microscope (AFM) (Fig. 1d, f) showed that both catalysts were composed of layered nanosheets with approximately equivalent thickness at around 20 nm. The two catalysts were examined by powder X-ray diffraction (PXRD). As shown in Fig. 1g, the XRD patterns of $KMnPO_4 \cdot H_2O$ were consistent with the calculated patterns from the reported crystal structure (Fig. 1g, top). The diffraction peaks of 2θ at 10.6°, 21.1°, and 32.4° can be indexed to (010), (011), and (121) crystal planes of $KMnPO_4 \cdot H_2O$. The

diffraction peaks of the dehydrated sample were also consistent with the patterns from the crystal structure of KMnPO₄ (Fig. 1g, bottom). The diffraction peaks of 2θ at 14.1°, 20.6°, and 28.4° can be indexed to (011), (020), and (12-1) crystal planes of KMnPO₄. The lattice fringes from the HR-TEM images (Supplementary Fig. 2) are also consistent with the XRD patterns. Therefore, it has been confirmed that the coordinated water molecules in $KMnPO_4 \cdot H_2O$ can be removed by heat treatment[37]. The thermogravimetric analysis (TGA) further confirmed the removal of coordinated water molecules in $KMnPO_4 \cdot H_2O$ (Fig. 1h). When the temperature reached 300 °C, the weight loss was 9.5 wt%, which was consistent with the removal of a molecule of water ligand. The Mn valences of the two catalysts were analyzed by X-ray photoelectron spectroscopy (XPS). The Mn 3s and Mn 2p have identical peak positions in the two catalysts (Fig. 1i, j). The peak position of the Mn $2p_{3/2}$ at 641.35 eV, the satellite peak at around 646.25 eV, and the peak splitting of 6.20 eV for the Mn 3s spectra indicated that the surface Mn valences were two in both catalysts[38,39]. The XPS spectra of P, O, and K elements were displayed in Supplementary Fig. 3. The two homologous catalysts have similar morphology and identical Mn valence, which meets the prerequisite for comparative study of the relationship between structure and catalytic performance.

### The crystal structures of the electrocatalysts

As shown in Fig. 2, the crystal structures of KMnPO₄ (ICSD-78840, a = 5.4813 Å, b = 8.6274 Å, c = 8.8865 Å, α = 87.7°, β = 89.1°, γ = 88.0°, P-1, Z = 4) and $KMnPO_4 \cdot H_2O$ (ICSD-71177, a = 5.6768 Å, b = 8.3358 Å, c = 4.9058 Å, α = 90.0°, β = 90.0°, γ = 90.0°, Pmn2₁, Z = 2) were defined[40,41]. The crystal structures of other planes are shown in Supplementary Fig. 4. In KMnPO₄, the crystal structure contains only one type of 4-coordinated Mn(II) centers (Fig. 2a), where all the four coordinated oxygen atoms are from the phosphate groups. The [MnO₄] tetrahedrons and phosphate tetrahedrons share via corners to form a layered structure with hexagonal rings along the b-axis. The K⁺ ions exist in the rings to stabilize the entire framework as counterbalance cations. At present, most manganese phosphates have been reported to be 6-coordinated or 5-coordinated. It is rare to find manganese phosphate with only 4-coordinated Mn(II) centers, which is

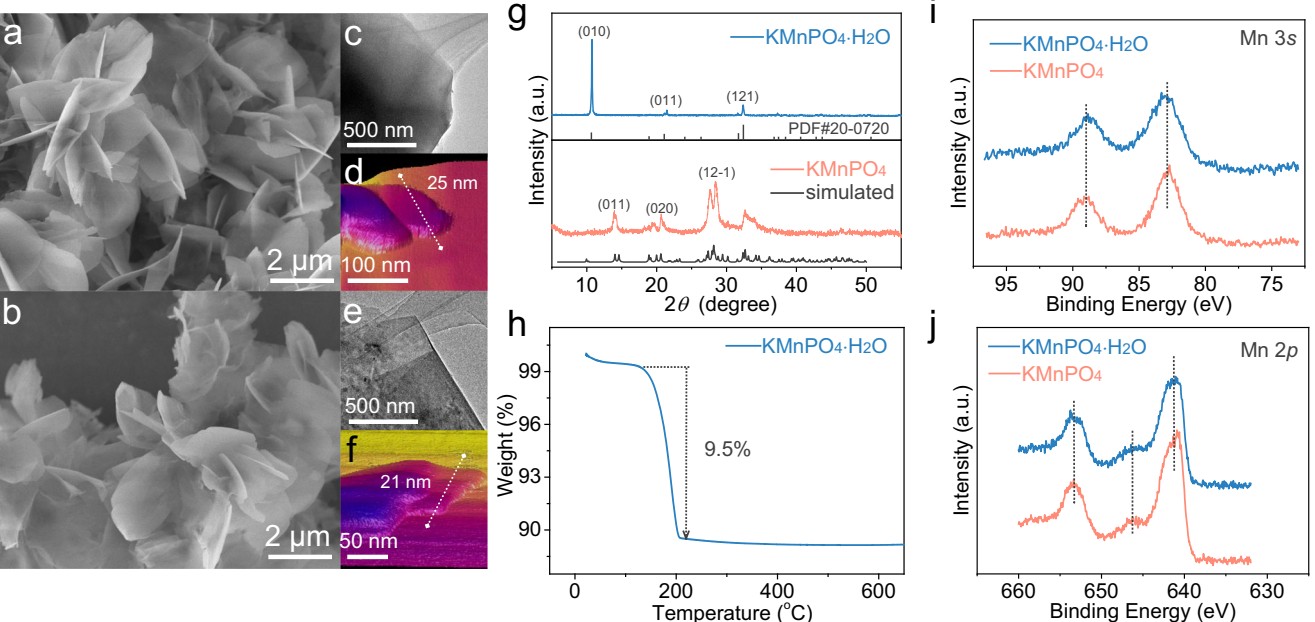

**Fig. 1 | Physical characterizations of the electrocatalysts. a**, **b** SEM, **c**, **e** TEM, and **d**, **f** AFM images of $KMnPO_4 \cdot H_2O$ (**a**, **c**, **d**) and KMnPO₄ (**b**, **e**, **f**). **g** PXRD patterns of $KMnPO_4 \cdot H_2O$ (top) and KMnPO₄ (bottom); black lines are simulated from crystal structures. **h** The TGA analysis of $KMnPO_4 \cdot H_2O$. **i**, **j** The Mn 3s and Mn 2p XPS spectra of the two catalysts.

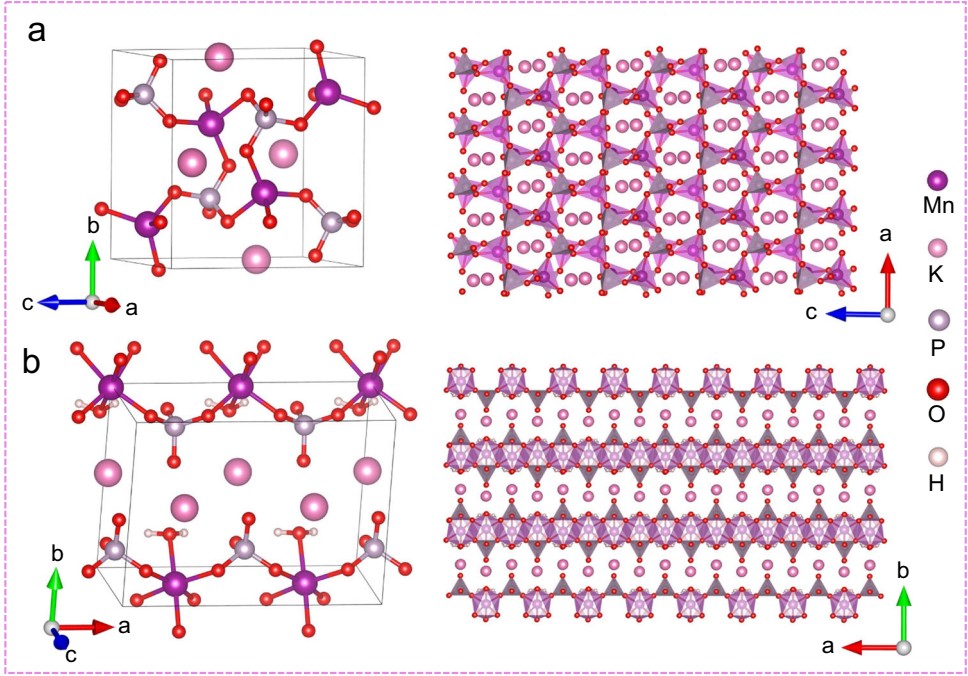

**Fig. 2 | Crystal structures of the electrocatalysts.** The crystal structures of (**a**) KMnPO$_4$ and (**b**) KMnPO$_4$•H$_2$O. The purple motifs are Mn octahedrons and Mn tetrahedrons; the gray motifs are phosphate tetrahedrons.

unique to study the OER mechanism. In KMnPO$_4$•H$_2$O, the crystal structure contains only 6-coordinated Mn(II) centers (Fig. 2b). The Mn atom is coordinated with six oxygen atoms to form an octahedron, in which five oxygen atoms are from the phosphate groups and another one oxygen atom is supplied by the coordinated water molecule. The [MnO$_6$] octahedron and phosphate tetrahedron also form a layer along the c-axis through corner-sharing. The K$^+$ ions exist as counterbalance cations to stabilize the entire framework between the layers. The same counterbalance cation in the two catalysts avoid the introducing of foreign cations, which are widely used to regulate the core structures of active centers in reported systems and might cause unexpected impact on the catalytic performance[42]. When a catalyst contains multiple Mn center structures, it is difficult to determine the real active site and establish the structure-activity relationship. In this work, the core Mn structure in the two catalysts is single and unambiguous, which provides an ideal platform for studying the OER mechanism.

### Electrocatalytic studies for water oxidation

The electrocatalytic OER performance of the manganese phosphates were explored by cyclic voltammetry (CV) in a 0.05 M pH = 6.99 PBS solution on a glass carbon (GC) electrode. In Fig. 3a, the KMnPO$_4$ exhibits higher catalytic activity than KMnPO$_4$•H$_2$O does. To reach a current density of 1.0 mA•cm$^{-2}$, the potentials required for KMnPO$_4$ and KMnPO$_4$•H$_2$O are 1.73 and 1.89 V vs reversible hydrogen electrode (RHE, all potentials are compared with RHE hereafter unless otherwise stated), respectively. In order to determine the intrinsic activity, the currents are normalized to the surface areas of the catalysts[43,44]. The electrochemical surface areas (ECSA) of KMnPO$_4$ and KMnPO$_4$•H$_2$O were compared based on their surface capacitances at 8.88 and 7.69 μF, respectively (Supplementary Fig. 5). Moreover, the Brunner-Emmett-Teller (BET) surface areas of KMnPO$_4$ and KMnPO$_4$•H$_2$O were measured to be 8.76 and 6.41 m$^2$•g$^{-1}$, respectively (Supplementary Fig. 6). The OER current densities based on ECSA and BET surface areas indicate the higher intrinsic activity from KMnPO$_4$ with 4-coordinated Mn centers. A four-point probe and static contact angle methods were used to test the physical conductivity (Supplementary Fig. 7) and hydrophilicity (Supplementary Fig. 8) of the two catalysts,

respectively. The conductivity and hydrophilicity are similar between the two catalysts, which avoids unfair comparison of their OER activities[45,46]. Moreover, the catalytic currents in phosphate electrolytes with different concentrations are recorded to investigate the reaction response to buffer (Supplementary Fig. 9). The OER currents remained basically unchanged, elucidating that proton transfer and the OER rate are not affected by the buffer concentration[47,48]. The precatalytic events of the two catalysts were compared by the differential pulse voltammetry (DPV) (Fig. 3b)[49]. At -1.25 V, the first precatalytic oxidation (Mn$^{II/III}$) occurred in both catalysts. Subsequently, the second precatalytic oxidation (Mn$^{III/IV}$) occurred in both catalysts at -1.50 V. Compared with CV, DPV can reduce background interference. The area integration of the oxidation peaks (color filling) in Fig. 3b demonstrates their close peak areas. However, the redox peak areas of the second oxidation events of the two catalysts differ greatly. In other words, the concentration of Mn(IV) intermediate species is inconsistent in the two catalysts. Further, the concentrations of Mn(III) intermediates in the two catalysts are compared by measuring the UV-vis absorption spectra of the Mn(III) pyrophosphate (pp) complex, which has a typical absorption at 258 nm (Fig. 3c)[32]. After electrolysis at 1.35 V for 2 h in a 20 mM pp solution, the solutions of both catalysts have significant absorption of Mn(III)-pp. However, the absorption area (color filling) of KMnPO$_4$ is 2.5 times that of KMnPO$_4$•H$_2$O, indicating that the former has more remained Mn(III) species against the Jahn-Teller distortion-caused disproportionation. When the potential is 1.35 V, surface Mn(II) sites will be completely oxidized to Mn(III). The oxidation peak areas of Mn$^{II/III}$ in the two catalysts are basically equal, indicating that surface Mn(II) concentrations are equal in the initial catalysts[50]. Obviously, the Mn(III) species can be effectively retained in KMnPO$_4$ but undergoes disproportionation in KMnPO$_4$•H$_2$O. The DPV curves of the two catalysts were also recorded in electrolytes with different pH values. As shown in Fig. 3d, e, the precatalytic oxidation events (Mn$^{II/III}$) are pH-dependent with proton-coupled electron transfer (PCET) features[51,52]. The $E_{pa}$(Mn$^{II/III}$) value of KMnPO$_4$ has a linear relationship with pH values at a slope of −61.5 mV•pH$^{-1}$, while the slope of KMnPO$_4$•H$_2$O is −59.4 mV•pH$^{-1}$ (Fig. 3f). A typical 1H$^+$/1e process occurs in the first oxidation of both catalysts[50,53].

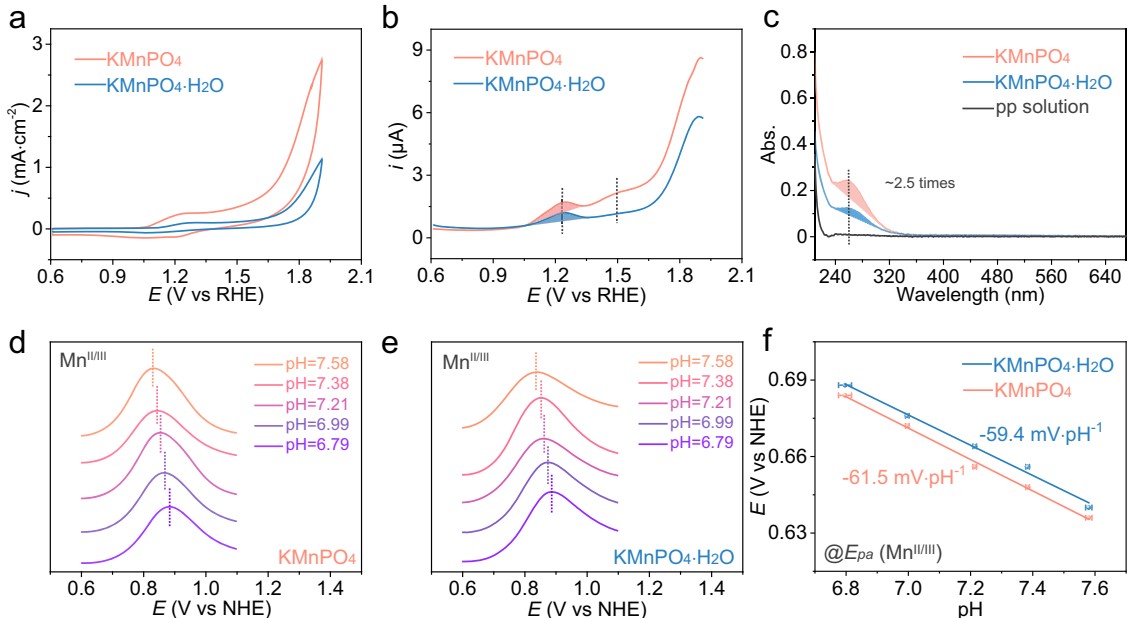

**Fig. 3 | Electrocatalytic OER measurements. a** CV and **b** DPV polarization curves of $KMnPO_4$ and $KMnPO_4 \cdot H_2O$ in 0.05 M PBS solution (pH = 6.99) (The solution resistance is 138.7 ± 1.2 Ω). **c** The UV-vis absorption spectra of the pyrophosphate solution after 2 h OER electrolysis at 1.35 V with $KMnPO_4$ and $KMnPO_4 \cdot H_2O$ (The solution resistance is 213.4 ± 3.2 Ω). The first precatalytic oxidation of $KMnPO_4$ (**d**) and $KMnPO_4 \cdot H_2O$ (**e**) from DPV in electrolytes with different pH values (The solution resistances are 145.6 ± 2.0 Ω (pH = 6.79), 138.7 ± 1.2 Ω (pH = 6.99), 136.6 ± 2.2 Ω (pH = 7.21), 135.9 ± 1.9 Ω (pH = 7.38), 133.0 ± 0.6 Ω (pH = 7.58), respectively). All standard deviations were obtained by three repeated measurements. **f** The potential responses of the two electrocatalysts to the pH values of the electrolyte at $E_{pa}$ ($Mn^{II/III}$). The error bars were the standard deviations of three repeated measurements.

Various ex-situ and in-situ techniques were employed to identify the structures of the intermediates of the two catalysts. The $Mn^{III/IV}$ oxidation is clearly observed from the DPV curves (Fig. 3b). The square wave voltammograms (SWV) in the cathodic direction after setting the electrodes at different potentials are shown in Supplementary Fig. 10 (the anodic scanning is also displayed for comparison)[54]. When the $KMnPO_4$ electrode was set at 1.80 or 1.90 V where water oxidation happens for 30 s and was subsequently scanned in the cathodic direction, two reduction peaks for the $Mn^{IV/III}$ and $Mn^{III/II}$ appeared. Decreasing the setting potentials for the $KMnPO_4$ electrode, a wide peak appeared in the $Mn^{III/II}$ reduction peak region and its area increased as the setting potential decreased. The phenomenon is because the formation of Mn(IV) or oxygen evolution happens at high potentials. The generation of Mn(IV) and oxygen consumes Mn(III) species. The $Mn^{III/IV}$ redox is quasi-reversible and water oxidation is irreversible in our case. Thus, when the electrode was set at higher potentials, the Mn(III) concentration would be lower to show a decreased cathodic peak of Mn(III) to Mn(II). In contrast, a broad reduction peak was found in the $KMnPO_4 \cdot H_2O$ electrode regardless of the setting potentials. These observations indicate that both catalysts can form Mn(IV) species at high potentials. The reduction of $Mn^{IV/III}$ is quasi-reversible and the concentration of Mn(IV) species is higher in the $KMnPO_4$ sample. To further understand the Mn(IV) intermediates with a $d^3$ electronic configuration, potential-dependent ex-situ electron paramagnetic resonance (EPR) measurements were conducted for the catalysts by freeze-quenching[55]. EPR spectroscopy is sensitive and can detect extremely low concentrations of Mn(IV) species. The catalysts on carbon cloth (CC) substrates after electrolysis at a certain potential were collected immediately and subsequently quenched in liquid nitrogen. The continuous-wave EPR spectra of the $KMnPO_4$ and $KMnPO_4 \cdot H_2O$ catalysts at various anodic potentials are shown in Fig. 4a, b. When the applied potential was 1.00 V or 1.20 V, no EPR signal was observed. As the applied potential increased to 1.40 V, the Mn(IV) signal near $g = 4.23$ appeared. When the applied potential further increased, the Mn(IV) signal gradually increased. The two catalysts

were also loaded on a transparent indium tin oxide (ITO) electrode to observe its in-situ UV-vis absorbance under different applied potentials[32,49,56]. As shown in Fig. 4c, d, the $KMnPO_4$ catalyst displayed increased absorbance of UV-vis at ~400 nm when the applied potential was increased from 1.50 V to 1.90 V. This broad peak is attributed to the d-d transitions of Mn(IV) or species with even higher Mn valence states[57]. In contrast, the $KMnPO_4 \cdot H_2O$ catalyst displays a much weaker absorbance signal of high-valent Mn species, indicating that the Mn(IV) concentration is low in the 6-coordination structure. In addition to the ex-situ EPR and in-situ UV-vis absorbance measurements, we performed in-situ Raman spectroscopy at different potentials (Fig. 4e, f) to gather further insights into the intermediates[58]. New signals appear in the region of 700 to 800 cm$^{-1}$, in which the vibration bands of high-valent Mn-oxo species are normally observed[59]. It is worth noting that a new broad peak centered at ~760 cm$^{-1}$ appeared as the applied potential increased to 1.40 V on the surface of $KMnPO_4$. This signal becomes more pronounced when the applied potential increases to 1.60 V. In addition, control experiments supports that the peak at 760 cm$^{-1}$ belongs to the Mn-oxo species generated during OER (Supplementary Fig. 11). According to previous studies on high-valent Mn-oxo species and our experimental results, it can be confirmed that the peak at ~760 cm$^{-1}$ belongs to Mn(IV)=O species[57,60]. A signal of Mn(IV) =O species can also be observed on the surface of $KMnPO_4 \cdot H_2O$ catalyst when the applied potential is increased to 1.60 V. In both catalysts, the signal of Mn(IV)=O species increased with the gradual increase of applied potential, which is consistent with the results of EPR and UV-vis. The Raman signals of the manganese phosphate structure does not change obviously with potential and time, which indicates the structural stability of the catalyst structure during catalysis.

The DPV curves near the $Mn^{III/IV}$ oxidation region of the two catalysts were recorded in electrolytes with different pH values (Supplementary Figs. 12 and 13)[61]. Like the $Mn^{II/III}$ oxidation, the precatalytic $Mn^{III/IV}$ oxidation events are also pH-dependent. The $E_{pa}(Mn^{III/IV})$ value of $KMnPO_4$ has a linear relationship with pH values at a slope of

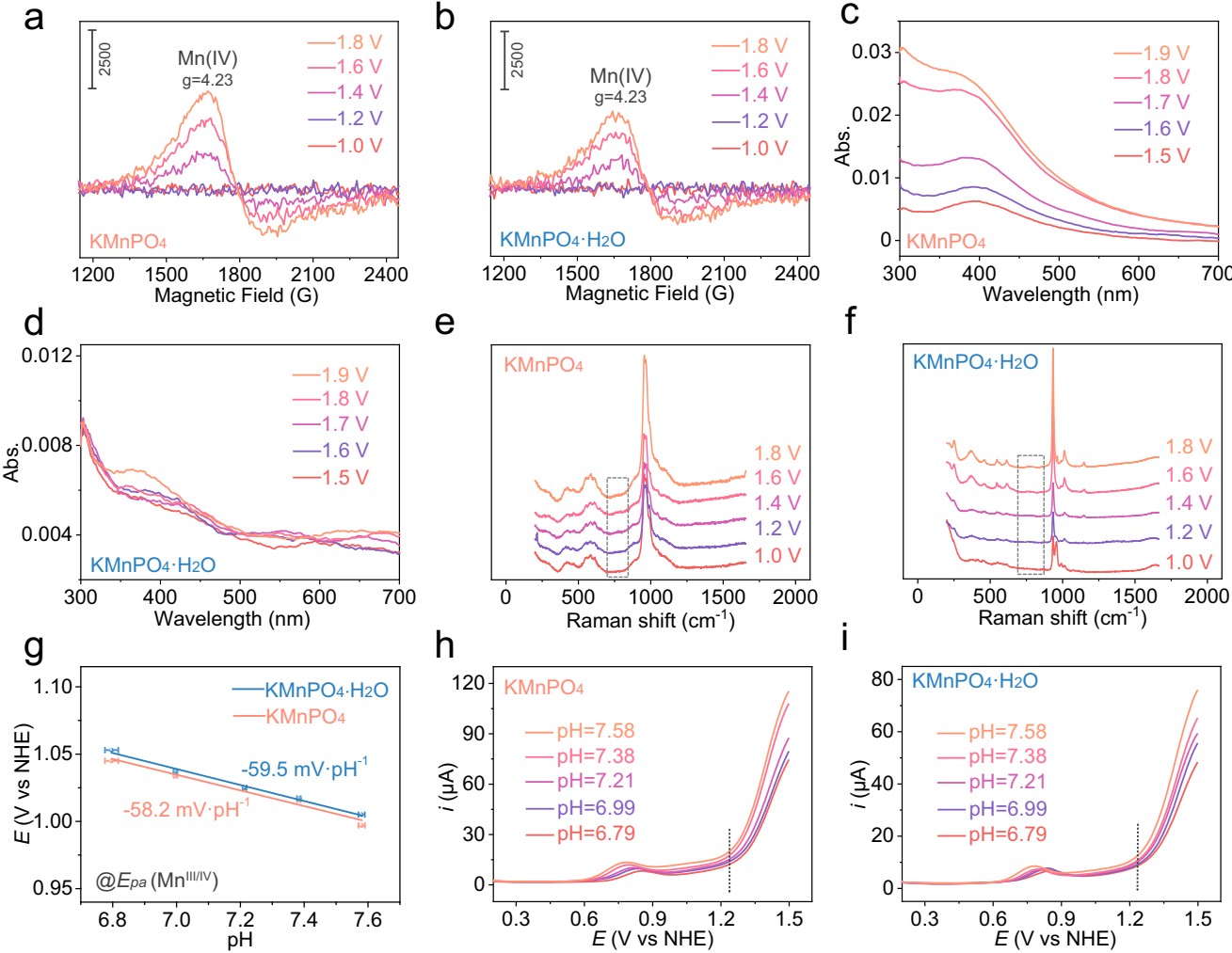

**Fig. 4 | Characterizations on the intermediates.** The ex-situ potential-dependent EPR spectra of KMnPO₄ (**a**) and KMnPO₄·H₂O (**b**). The in-situ potential-dependent UV-vis (**c**, **d**) and in-situ Raman (**e**, **f**) spectra of KMnPO₄ (**c**, **e**) and KMnPO₄·H₂O (**d**, **f**) (The solution resistance is 138.7 ± 1.2 Ω). **g** The potential responses of the two electrocatalysts to the pH values of the electrolyte at $E_{pa}$ (Mn^{III/IV}). The error bars were the standard deviations of three repeated measurements. The electrochemical responses of KMnPO₄ (**h**) and KMnPO₄·H₂O (**i**) in electrolytes with different pH values. (The solution resistances are 145.6 ± 2.0 Ω (pH = 6.79), 138.7 ± 1.2 Ω (pH = 6.99), 136.6 ± 2.2 Ω (pH = 7.21), 135.9 ± 1.9 Ω (pH = 7.38), 133.0 ± 0.6 Ω (pH = 7.58), respectively). All standard deviations were obtained by three repeated measurements.

−58.2 mV·pH⁻¹, while the slope of KMnPO₄·H₂O is −59.5 mV·pH⁻¹ (Fig. 4g). The results show that the second oxidation of the two catalysts also occurred as 1H⁺/1e process. On the one hand, the Mn^{II/III} and Mn^{III/IV} couples shift by roughly 59 mV·pH⁻¹, illustrating stepwise PCET oxidations of the active site in a typical Mn(II)−OH₂ to Mn(III)−OH and then to Mn(IV)=O pathway. On the other hand, the two catalysts began to catalyze in different pH buffer solutions after a potential of about 1.25 V (Fig. 4h, i), illustrating that the further oxidation of Mn(IV)=O, if necessary before the water oxidation happens, does not involve the transfer of protons[62]. The catalytic currents are dependent on the pH values of the electrolyte (Supplementary Fig. 14). As shown in Supplementary Fig. 15, the required potentials are proportional to the pH values with a linear slope of −95.7 and −110.8 mV·pH⁻¹ at $i$ = 45 μA for KMnPO₄ and KMnPO₄·H₂O, respectively. At different regions, the pH dependences of catalytic currents are slightly different. This phenomenon is because that these kinetic currents are probably overlapped with the Mn(IV) to Mn(V) oxidation currents, which are independent of pH. The Tafel slopes are determined to be 245 and 267 mV·dec⁻¹[63,64]. The reasons for the high Tafel slopes might be that (1) the oxidation currents of Mn(IV) to Mn(V) interfered the OER currents and (2) the formation of Mn−OOH is a pure chemical step instead of an electrochemical step. However, the similar kinetic values indicate parallel reaction pathways of the two catalysts.

As the formation Mn(V)=O from the oxidation of Mn(IV)=O can be concealed by water oxidation in aqueous solutions, the redox properties of KMnPO₄ and KMnPO₄·H₂O have also been investigated by DPV in organic CH₃CN/0.05M n-Bu₄NPF₆ solutions (Fig. 5a, c)[62,65]. Three successive oxidations are observed in the two catalysts, which are assigned to Mn^{II/III}, Mn^{III/IV}, and Mn^{IV/V} oxidations. The areas of Mn^{II/III} oxidation peaks are basically identical in the two catalysts, indicating that the number of Mn(II) sites on the two catalysts surfaces is close. This is consistent with the DPV results in aqueous solutions. The oxidation peak area of Mn^{III/IV} in KMnPO₄ is larger than that of KMnPO₄·H₂O, indicating that more active Mn(III) intermediate is retained in KMnPO₄. Similarly, the oxidation peak area of Mn^{IV/V} in KMnPO₄ is also larger than that of KMnPO₄·H₂O, indicating that more active Mn(IV) intermediate is generated in KMnPO₄ to form more Mn(V) species, which is the probable reason for its enhanced activity. The Mn^{IV/V} oxidation observed in CH₃CN/0.05M n-Bu₄NPF₆ was replaced with a strong catalytic current when 2 vol% water was added (Fig. 5b, d)[65]. Under such circumstances, the Mn^{III/IV} oxidation peaks in the two catalysts still exist. These results indicate that Mn(V) is

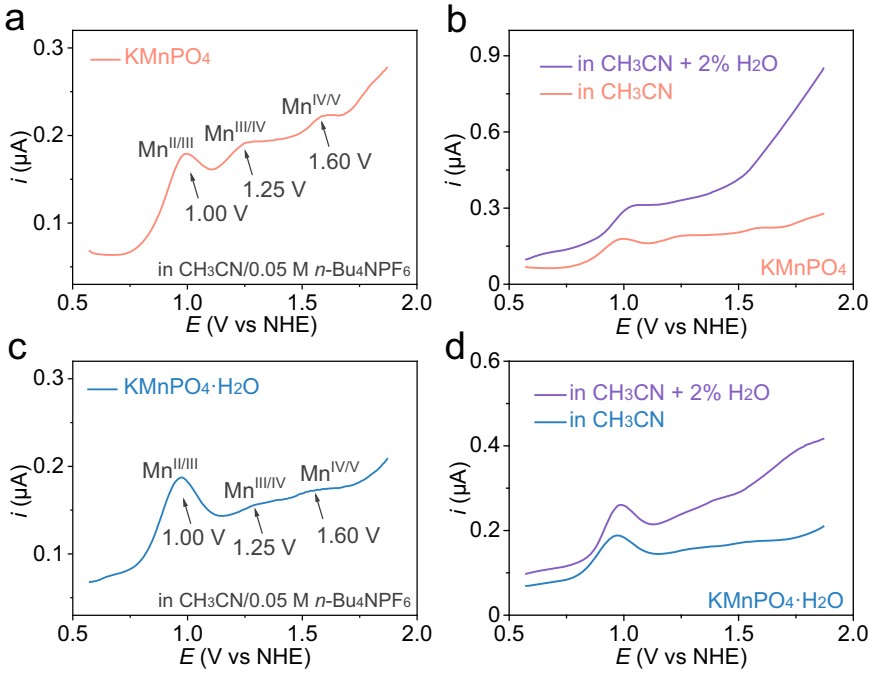

**Fig. 5 | The electrochemical behaviors in nonaqueous solutions.** The DPV curves at an amplitude of 5 mV for KMnPO$_4$ (**a**, **b**) and KMnPO$_4$•H$_2$O (**c**, **d**) in CH$_3$CN/0.05 M $n$-Bu$_4$NPF$_6$ with or without 2 vol% water, the solution resistance is 37.1 ± 2.8 Ω.

involved as the active species in the water oxidation to oxygen process. In addition, the absorption peak at approximately 400 nm does not change with time, which is sufficient to prove that Mn(IV)=O is at steady state and is at equilibrium with Mn(V)=O (Supplementary Fig. 16). The Mn(V)=O species has been detected and identified as active species for water oxidation in several reported Mn-complex homogeneous catalysts[60,62,66]. The determining of Mn(V)=O in heterogeneous catalysts for water oxidation will greatly increase the understanding of the mechanism of heterogeneous electrocatalytic water oxidation. The oxidation of Mn(IV)=O can also generate a Mn(IV)−O• radical. However, the Mn(IV)−O• and Mn(V)=O are two valence tautomeric forms at equilibrium[11]. Herein, we described the Mn(V)=O in a formal oxidation state manner. The stabilities of the two catalysts are a prerequisite for the study of OER mechanism. The stability of the catalysts is evaluated using controlled potential electrolysis without iR compensation within 12 h (Supplementary Fig. 17). The Raman and XPS spectra of the two catalysts after electrolysis are displayed in Supplementary Figs. 18 and 19. The Raman analysis confirms that the structures of the two catalysts are unchanged after electrolysis. The peak positions of the Mn 3$s$ and Mn 2$p$ spectra are unchanged, indicating the stability of the catalyst structure.

### Evaluation of water oxidation mechanisms

To explore the coordination environments and OER mechanisms of the two catalysts at atomic level, X-ray absorption near-edge structure (XANES) and extended X-ray absorption fine structure (EXAFS) spectra for the Mn K-edge are studied[57,67–69]. For 3$d$ transition metal-based catalyst, the pre-edge peaks are attributed to the forbidden 1$s$ to 3$d$ transition. The pre-edge peak intensity change is enunciative of the changes in the metal site proportion in octahedral and tetrahedral symmetry[70,71]. In short, the octahedral signal is wider and less intense and the tetrahedral signal is narrower and more intense. This is greatly because tetrahedral symmetry structure is highly non-centrosymmetric, which makes the $p$ to $d$ transition helpful for pre-edge peaks. In KMnPO$_4$ catalyst, Mn site is [MnO$_4$] tetrahedral structure, the pre-edge peak signal is more obvious than that of [MnO$_6$] octahedral structure of KMnPO$_4$•H$_2$O catalyst. The position of the MnO absorption edge

could be used as an indicator of valence states. Supplementary Fig. 20a suggests that the valence state of Mn ions is two in the two catalysts. According to the EXAFS fitting parameters (Supplementary Fig. 20b, c), the Mn−O average coordination numbers (N) were calculated to be 3.872 for KMnPO$_4$ and 5.950 for KMnPO$_4$•H$_2$O. The fitting results were consistent with the crystal structure data. The XANES spectra show that the Mn K- edge energies of KMnPO$_4$ and KMnPO$_4$•H$_2$O increase with the increased applied potential from 1.00 V to 1.80 V, indicating that the Mn oxidation state lifts with the increase of applied potential (Fig. 6a, b)[72–74]. In addition, a stable structure is a necessary prerequisite for studying the mechanism. The almost overlapped Mn K-edge oscillation curves (Fig. 6c, d) indicate the similar geometric structure of the catalysts at different potentials. The Fourier-Transformed (FT) magnitude plots of the Mn K-edge EXFAS spectra of KMnPO$_4$ at different potentials are shown in Fig. 6e, where the wide signal at -1.5 Å can be attributed to the Mn−O scattering path in the [MnO$_4$] group and Mn−O$_w$ (adsorbed water around the site) scattering path at higher radial distance. The FT spectra were not phase-corrected, and the bond length would be shorter than the real value. For KMnPO$_4$, the first shell is mainly a wide signal at 1.00 V which can be attributed to the Mn−O scattering path in the [MnO$_4$] group and Mn−O$_w$ scattering path (O$_w$, adsorbed water around the site). When the potential rises to 1.20 V, the first shell completely changes from a wide signal to a peak, indicating that the adsorbed water molecules have bonded to the active site (or other species with weak Mn−O bond) to probably form [MnO$_5$] groups with entirely Mn−O scattering path. At 1.40 V, the broad peak of the first shell becomes three weak peaks, among which the signal at -1.2 Å can be attributed to the Mn=O scattering path, the signal at -1.5 Å is still the Mn−O scattering path, and the signal peak at -1.8 Å is the Mn−O$_w$ scattering path (or from species with weak Mn−O bond). The oxygen in Mn−O is mainly from phosphate, while the oxygen in Mn=O is an absolute oxygen atom with more ionic bond character. Therefore, it is clear that Mn(III) can be oxidized to Mn(IV)=O species when the potential is up to 1.40 V. The signal at 1.60 V is basically identical as that at 1.40 V. The slight change in the distance of the scattering path might be due to the formation of Mn(V) =O and corresponding species from O−O bond formation. When the

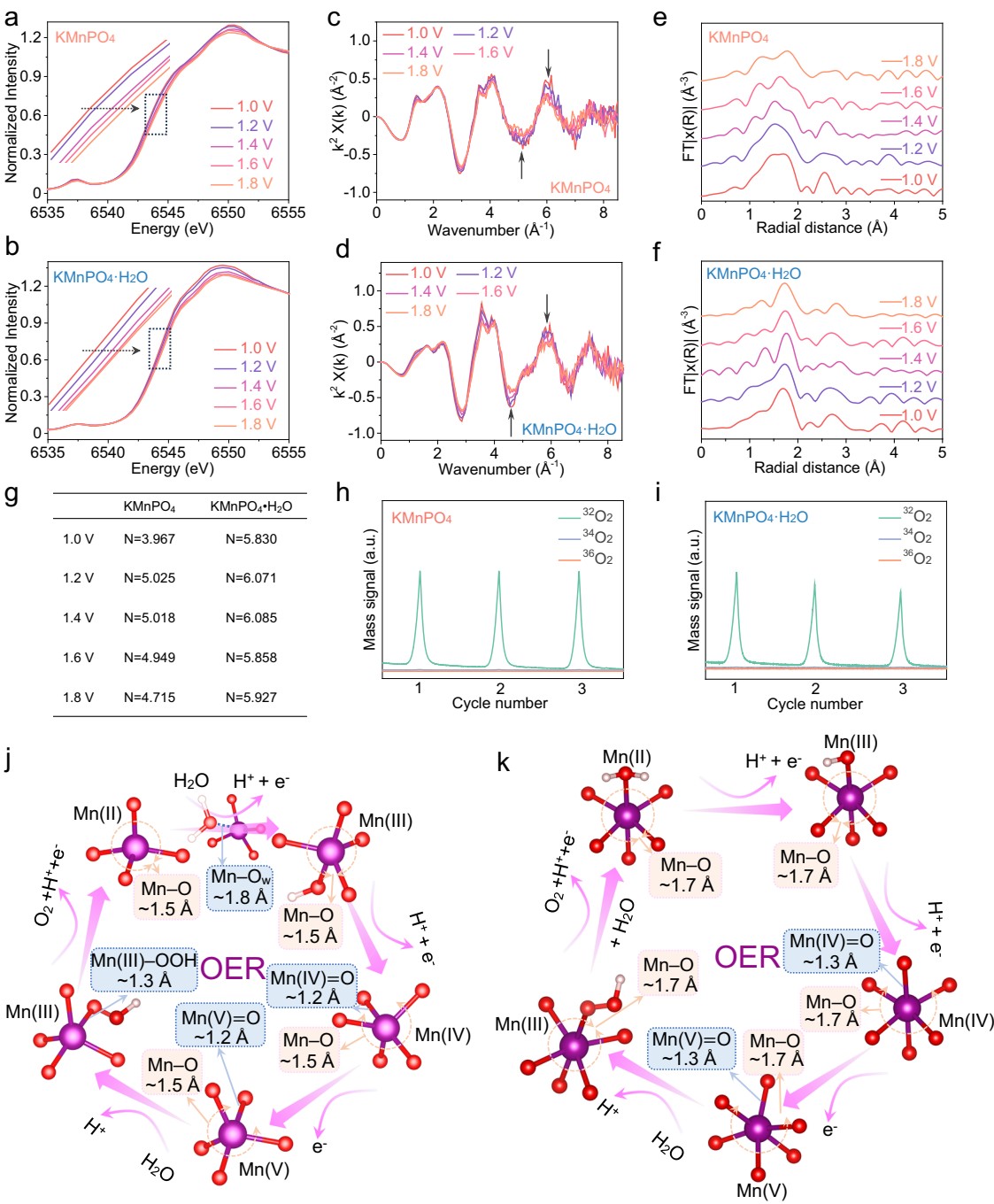

**Fig. 6 | OER catalytic mechanism analyses. a, b** In-situ XANES spectra for Mn K-edge of KMnPO₄ and KMnPO₄·H₂O at potentials ascending from 1.00 to 1.80 V in 0.05 M PBS (pH=6.99) (The solution resistance is 138.7 ± 1.2 Ω). **c, d** The EXAFS oscillation functions of KMnPO₄ and KMnPO₄·H₂O. **e, f** Mn K-edge EXAFS spectra for KMnPO₄ and KMnPO₄·H₂O at different potentials. The Fourier transform (FT) spectra were not phase-corrected. **g** The coordination number changes of KMnPO₄ and KMnPO₄·H₂O during OER. **h, i** DEMS signals of $^{36}O_2$ ($^{18}O^{18}O$, mass/charge ratio ($m/z$ = 36), $^{34}O_2$ ($^{16}O^{18}O$, $m/z$ = 34), and $^{32}O_2$ ($^{16}O^{16}O$, $m/z$ = 32) from the gaseous products for KMnPO₄ (**h**) and KMnPO₄·H₂O (**i**) catalysts in $H_2^{18}O$ aqueous PBS electrolyte during three times of cycles in the potential range of 0.90 to 2.10 V versus RHE at a scan rate of 5 mV·s⁻¹ (The solution resistance is 138.7 ± 1.2 Ω). Proposed OER mechanisms for KMnPO₄ (**j**) and KMnPO₄·H₂O (**k**) in neutral PBS solutions.

potential rises to 1.80 V, the three signal peaks of the first shell layer become two signal peaks. At 1.80 V, the O₂ is formed and released. A variety of adsorbents (substrates, intermediates and products) exist in equilibrium with the continuous release of O₂. The re-adsorption of substrate water molecules also happens for any refreshed catalytic site. These adduct together lead to the right shift of the observed signal peak. This right shifted peak centered at the position close to the right shoulder of the broad peak from the sample at 1.00 V, which is

consistent with that the catalysis of O₂ formation is completed and the catalytic Mn center goes back to its original low valence state. For the KMnPO₄·H₂O catalyst (Fig. 6f), the first shell is mainly an independent signal peak at ~1.7 Å which can be attributed to the Mn−O scattering path in the [MnO₆] group. At 1.00 V and 1.20 V, there is only one signal peak, indicating that the average bond length of Mn−O does not change significantly during the $Mn^{II/III}$ oxidation. The main reason is that the coordinated water molecule, not the adsorbed water

molecule, is involved in this process. When the potential increased to 1.40 V or higher, a new scattering path appeared at ~1.3 Å which could be attributed to the Mn(IV)=O species. Overall, the Mn structure in the 6-coordinated $KMnPO_4 \cdot H_2O$ is more rigid than in the 4-coordinated $KMnPO_4$. In order to accurately determine the Mn coordination structure of the two catalysts at different potentials, the quantitative EXAFS curve fitting analysis in R-space was conducted at the Mn K-edge (Supplementary Figs. 21 and 22). According to the EXAFS fitting parameters, the Mn−O average coordination numbers of $KMnPO_4$ at different potentials were calculated as 3.967 (at 1.00 V), 5.025 (at 1.20 V), 5.018 (at 1.40 V), 4.949 (at 1.60 V) and 4.715 (at 1.80 V), which can reveal the coordination number changes of $KMnPO_4$ during OER (Fig. 6g). For the $KMnPO_4 \cdot H_2O$ catalyst, the Mn−O average coordination numbers at different potentials were calculated as 5.830 (at 1.00 V), 6.071 (at 1.20 V), 6.085 (at 1.40 V), 5.858 (at 1.60 V) and 5.927 (at 1.80 V), indicating that the Mn octahedral configuration did not change during OER. Additionally, the Wavelet Transforming (WT) of EXAFS oscillations provided a higher resolution of the structure in R spaces (Supplementary Figs. 23 and 24). The WT contour plots of $KMnPO_4$ and $KMnPO_4 \cdot H_2O$ have only one intensity maximum at 3.7 $A^{-1}$ and 3.8 $A^{-1}$ in R space, corresponding to the Mn−O coordination. The position of the maximum intensity shifts slightly with the change in potential, which is consistent with the above R-space analysis results. To further elucidate the OER mechanisms that occur on the $KMnPO_4$ and $KMnPO_4 \cdot H_2O$ catalysts, isotope-labeled in-situ differential electrochemical mass spectroscopy (DEMS) measurements were accomplished using 0.05 M PBS $H_2^{18}O$ or $H_2^{16}O$ electrolyte (Fig. 6h, i and Supplementary Fig. 25)[75,76]. First, $KMnPO_4$ and $KMnPO_4 \cdot H_2O$ were loaded on the porous Au film and labeled with $^{18}O$ by performing three LSV cycles (0.90 to 2.10 V versus RHE) in the 0.05 M PBS $H_2^{18}O$ electrolyte. Then, the $^{18}O$-labeled catalysts were washed with a large amount of water to remove the adsorbed $H_2^{18}O$, followed by three cycles in the 0.05 M PBS $H_2^{16}O$ electrolyte. The signals detected by the two catalysts in 0.05 M PBS $H_2^{18}O$ electrolyte are mainly the mass spectrum current generated by $^{36}O_2$. The results show that the conventional adsorbate evolution mechanism (AEM) is dominant in the two catalysts. At the same time, a weak $^{34}O_2$ current signal was also detected, in which the content ratio of $^{34}O_2$ to $^{36}O_2$ in $KMnPO_4$ catalyst was 5.1%, while the content ratio of $^{34}O_2$ to $^{36}O_2$ in $KMnPO_4 \cdot H_2O$ catalyst was 9.5%, further indicating that the coordination water of $H_2^{16}O$ in the 6-coordinate structure participated in the water oxidation process. All the signals obtained by LSV cycling in 0.05 M PBS $H_2^{16}O$ electrolyte after labeling were $^{32}O_2$ signals, and no $^{36}O_2$ signals were observed.

Based on the results of electrochemical and spectral studies, the OER mechanisms for the two catalysts are proposed. OER mechanism comparison for $KMnPO_4$ and $KMnPO_4 \cdot H_2O$ is shown in Fig. 6j, k. Generally, Mn atom is favorably coordinated in the form of $[MnO_6]$ in inorganic structures, while the form of $[MnO_4]$ is rare. All the Mn atoms in $KMnPO_4$ were 4-coordinated. Firstly, a solvent water molecule is adsorbed on the Mn site with the structural change from $[MnO_4]$ to $[MnO_5]$. Then, two successive PCET oxidations of Mn(II)−$OH_2$ to Mn(III)−OH and then to Mn(IV)=O happen. The $[MnO_5]$ is a typical trigonal bipyramidal structure that can maximize the stability of Mn(III) intermediates and improve catalytic performance. We calculated the average lattice distortion index of Mn atom of $[MnO_5]$ in $KMnPO_4$ and $[MnO_6]$ in $KMnPO_4 \cdot H_2O$[27]. The index of the former (0.0487) is higher than that of the latter (0.0426). The structure of $[MnO_5]$ in $KMnPO_4$ is thermodynamically favored in retaining the generated Mn(III)−OH. Secondly, the Mn(IV)=O occurs an electron transfer oxidation to Mn(V)=O active species. Then, another substrate $H_2O$ nucleophilic attack Mn(V)=O active species to give Mn(III)−OOH and then to release $O_2$. For the $KMnPO_4 \cdot H_2O$ catalyst with the traditional 6-coordination structure, there is a coordination water molecule in the $[MnO_6]$ structure. As the potential rises, the coordination $H_2O$

loses a proton to OH with the metal center losing an electron. Similarly, two successive PCET oxidations of Mn(II)−$OH_2$ to Mn(III)−OH and then to Mn(IV)=O happen, and the structure is always maintained as $[MnO_6]$ structure. The Mn(IV)=O species is oxidized to the Mn(V)=O species by single electron transfer, and the Mn(V)=O species reacts with $H_2O$ to form Mn(III)−OOH which further releases $O_2$. By comparing the OER mechanisms of $[MnO_4]$ and $[MnO_6]$ structures, the following conclusions can be drawn. On the one hand, the high activity of $KMnPO_4$ is mainly due to the structural transformation of $[MnO_4]$ to $[MnO_5]$ during the first PCET process, which retains more Mn(III) active intermediates. On the other hand, regardless of the $[MnO_4]$ or $[MnO_6]$ site structures, their active species are all high valent Mn(V)=O species.

To understand the catalytic activity and the mechanism of catalysts with respect to its local structure on an atomistic level, we performed first principles density functional theory (DFT) calculations. In specific, all Mn sites in $KMnPO_4$ are tetrahedral configuration of $[MnO_4]$ group. Therefore, the $[MnO_4]$ surface and a separate $H_2O$ is selected as the computational model of $KMnPO_4$. As shown in Fig. 7a, water molecules are first adsorbed on the surface Mn site to form Mn(II)−$H_2O^*$, and then dissociated into a proton and Mn(III)−$OH^*$, which splits again into a proton and Mn(IV)−$O^*$. The Mn(IV)−$O^*$ will undergo further oxidation and structural change to Mn(V)−$O^*$. The oxygen atoms on Mn(V)−$O^*$ recombine with the water molecules to form the $OOH^*$ bond, eventually forming the $O_2$ gas. As shown in Fig. 7b, the coordinated water molecules at the Mn site on the surface of the $KMnPO_4 \cdot H_2O$ catalyst can generate $O^*$ species through two successive PCET processes. The subsequent OER steps are consistent with the process of $KMnPO_4$ catalyst. The Bader charges of the Mn−O structure for $KMnPO_4$ and $KMnPO_4 \cdot H_2O$ catalysts are −1.452 and −1.488, respectively ("−" symbol represents the loss of electrons by the atom) (Supplementary Tables 1–2). The proximity of the Bader charges of the two catalysts indicates that the valence states of Mn sites are similar. We calculated the Gibbs free energy of each step in $KMnPO_4$ catalyst (Fig. 7c). The water molecules adsorbed on the Mn active site will form a twisted triangular bipyramid structure, which is easy to lose a proton. Compared with the easy generation of $OH^*$ (0.94 eV), the generation of $O^*$ requires a higher energy change (1.58 eV) for structural transformation. This high energy change for Mn(III) to Mn(IV) indicates that it is important to retain the Mn(III) intermediate species that is prone to disproportionation in OER. With further oxidation of the Mn site, the $[MnO_5]$ structure undergoes another conformational change. It is worth noting that the highest energy change (1.66 eV) is required to generate $OOH^*$, indicating that this process might be a rate limiting step in $KMnPO_4$ catalyzed water oxidation. As the O−O bond formation is a slow chemical step after a fast electrochemical step (Mn(IV)=O to Mn(V)=O), it is also reasonable to infer that the Mn(V)=O species is at equilibrium with Mn(IV)=O, the concentration of which determines the intrinsic activity of water oxidation. Once $OOH^*$ species is generated, it is easy to release $O_2$ gas (0.95 eV). In these redox steps, the 5-coordination geometric structure remains unchanged, but the atomic position undergoes changes, indicating that the $[MnO_4]$ structure is very flexible in the process of water oxidation. This is the main advantage of the $[MnO_4]$ structure in water oxidation, as it is thermodynamically favored in retaining Mn(III)−OH and generating Mn(V)=O. In $KMnPO_4 \cdot H_2O$ catalyst, the $\Delta G$ values of OER steps are 0.57 eV (Mn(II)−$OH_2$ to Mn(III)−OH), 1.58 eV (Mn(III)−OH to Mn(IV)−O), 1.91 eV (Mn(V)−O to Mn(III)−OOH) and 1.09 eV (Mn(III)−OOH to $O_2$) (Fig. 7d). Consistent with the $KMnPO_4$ catalyst, the generation of $O^*$ requires a relatively high energy change (1.58 eV). The formation of $OOH^*$ requires the highest energy change of 1.91 eV, which is higher than that of $KMnPO_4$ (1.66 eV). The OER is thermodynamically favored in $KMnPO_4$, as the free energy changes are more evenly distributed in

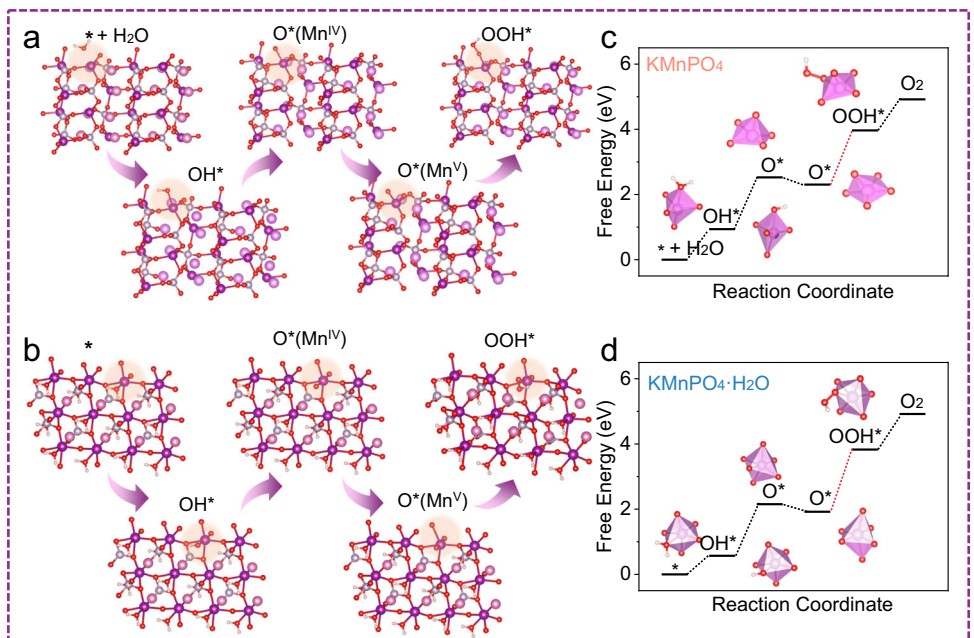

**Fig. 7 | Evaluation of the catalytic mechanism by DFT calculations.** The calculated OER pathways of the $KMnPO_4$ (**a**) and $KMnPO_4 \cdot H_2O$ (**b**) catalysts. Schematic of the Gibbs free energy changes for the elementary steps during OER based on DFT calculations in $KMnPO_4$ (**c**) and $KMnPO_4 \cdot H_2O$ (**d**) catalysts.

$KMnPO_4$ catalyst. Through the comparative study of the two catalysts, it is further shown that the rate-limiting step of water oxidation in the manganese phosphate system is the oxygen-oxygen bonding step.

In summary, we synthesized an unusual manganese phosphate with 4-coordinated Mn centers ($KMnPO_4$) from the calcination of a 6-coordinated manganese phosphate monohydrate ($KMnPO_4 \cdot H_2O$). The two materials constitute an ideal platform to study the structure-performance correlation in electrocatalytic water oxidation, as the two catalysts share many physical similarities except their distinct Mn coordination structures. The 4-coordinated $KMnPO_4$ exhibited much higher water oxidation activity in a neutral solution. The two catalysts exhibited similar precatalytic oxidation events. Two successive PCET oxidations of $Mn(II)-OH_2$ to $Mn(III)-OH$ and then to $Mn(IV)=O$ happened before an electron transfer oxidation of $Mn(IV)=O$ to $Mn(V)=O$. The presence of $Mn(III)$, $Mn(IV)$, and $Mn(V)$ active species were elucidated in the two catalysts during water oxidation. The $Mn(V)=O$ species is demonstrated to be the substance for O–O bond formation in water oxidation. Combined with XAS results and theoretical studies, the coordination environment of the Mn center during water oxidation was determined. In $KMnPO_4 \cdot H_2O$, the Mn–S scattering path and Mn coordination number did not change significantly, as the originally coordinated water in the $[MnO_6]$ group involved in the whole process of water oxidation. In $KMnPO_4$, the Mn coordination structure changed at an early stage from 4-coordinated $[MnO_4]$ to 5-coordinated $[MnO_5]$ motif with the additional oxygen supplied from an adsorbed water molecule around the Mn site. The $[MnO_5]$ motif displays a triangular biconical configuration. Later, the 5-coordination geometric structure remained unaltered but the atomic position underwent changes during water oxidation. The structure flexibility of $[MnO_5]$ is thermodynamically favored in retaining $Mn(III)-OH$ and generating $Mn(V)=O$. The $Mn(V)=O$ species is at equilibrium with $Mn(IV)=O$, the concentration of which determines the intrinsic activity of water oxidation. On the basis of the detailed comparative studies, the mechanism of Mn-based heterogeneous electrocatalyst for water oxidation can be logically and legibly understood.

## Methods

### Synthesis of materials
In a typical process, 0.5 mmol $MnCl_2 \cdot 4H_2O$ and 5.6 mmol $K_2HPO_4$ were dissolved in 5 mL of acetone and 15 mL of water stirred at room temperature for 24 h. The obtained potassium manganese phosphate monohydrate ($KMnPO_4 \cdot H_2O$) was collected by centrifugal washing with water three times and dried in oven at 60 °C. The obtained white solid was further heat treated to prepare potassium manganese phosphate ($KMnPO_4$) by calcination at 300 °C for 1 h in muffle furnace.

### Physical characterizations
Powder X-ray diffraction (PXRD) data were gained on Rigaku D/Max2550VB + /PC X-ray diffractometer at 40 kV and 100 mA using Cu Kα radiation with λ = 1.5406 Å. The scanning electron microscope (SEM) images were obtained on a Hitachi SU8020 cold-emission field emission scanning electron microscope (the accelerating voltage = 5 kV). The transmission electron microscopy (TEM) images were collected using a FEI Tecnai G2 F20 field TEM operated with 200 kV. The experimental samples were dispersed by water and then loaded on carbon-coated copper grids for analysis. The atomic force microscope (AFM) images were obtained on a Bruker Dimension ICON instrument. The experimental samples were dispersed with water and loaded onto a 1 cm × 1 cm silicon wafer for AFM analysis. The X-ray photoelectron spectroscopy (XPS) analysis was gained on a Kratos AXIS ULTRA XPS analyzer using monochromatic Al Kα X-ray source with hν = 1486.6 eV. The binding energy correction was carried out using C 1s peak (284.6 eV) arising from the adventitious hydrocarbon. Thermal analysis was put into practice by heating the dry solid powder samples at a rate of 5 °C/min under nitrogen flow at 100 mL/min over 25 °C to 800 °C in a TGA Instruments SDT Q600. The physical adsorption surface area was obtained by Brunauer-Emmett-Teller method (BET). The UV-vis absorption spectra of the supernatant of solid samples electrolyzed in pyrophosphate solution for 2 h were detected in the range of 200–800 nm using Shimadzu UV 3600 under room conditions. In-situ UV-vis absorption spectra measurements were performed on the same equipment (Supplementary Fig. 26). The In-situ Raman spectra of samples were conducted in a HORIBA LabRAM Odyssey with a 30 mW He/Ne laser at 532 nm laser (Supplementary Fig. 27). The

catalysts were drop-casted onto the ITO electrode (1 cm²) at a loading of 2 mg•cm⁻² for Raman spectra recording (laser intensity: 25%, spectral collection time: 60 s). Before Raman measurement, the electrocatalyst was allowed to be stabilized for 5 min at given potentials. The applied potential was kept constant during the Raman measurement. Sheet conductivities were measured by four points probe resistivity tester (SZJGDZ). The contact angle images of two samples were obtained on a video-based contact angle measurement by Dataphysics (OCA 20). The in-situ isotope-labeled DEMS measurements were collected on an operando system by LingLu Instruments (Supplementary Fig. 28). The system consists of a mass spectrometer with a high vacuum chamber and another chamber connected to an electrochemical cell. The generated oxygen products are directly transferred into the vacuum chamber for mass spectrometer analysis. In the electrochemical cell, Ag/AgCl and Pt are served as reference and counter electrodes, respectively. The Au film supported on the porous polytetrafluoroethylene membrane is used as the working electrode (0.785 cm², 1.27 mg•cm⁻²). All EPR measurements were carried out at a Germany-BROCK-A300 instrument. The catalysts were drop-casted onto the carbon cloth electrode (0.5 cm²) at a loading of 2 mg•cm⁻² for EPR signal recording. Three scans were collective for each spectrum. The electrolysis was directed using CH Instruments (CHI 600D) at pH = 6.99 in 0.05 M PBS. The set potentials were applied to each sample for 5 min. After 5 min of electrolysis, the carbon cloth electrode was quickly transferred to the EPR tube. The EPR tube is then immediately frozen and stored in 77 K liquid nitrogen. Mn K-edge X-ray absorption spectra (XAS) were collected at the BL14W beamline from Shanghai Synchrotron Radiation Facility. Electrocatalysts for operando XAS measurements were analyzed in an in-situ electrolytic cell at room temperature (Supplementary Fig. 29). The working electrode was prepared by loading two catalysts on a carbon cloth (2 cm²) with a load of 2 mg•cm⁻². Before collecting XAS spectra, the electrocatalyst was electrolytically stabilized at a given potential for 5 min. During XAS measurement, the electrolytic potential remains constant. The XANES and EXAFS data were analyzed and fitted using ATHENA and ARTEMIS software programs.

## Electrochemical studies

All electrochemical experiments were carried out using the Electrochemical Workstation of CH instrument (CHI 660E) at room temperature. Standard three electrode system procedure is followed: glass carbon (GC) (0.07 cm²) electrode as working electrode, graphite rod as auxiliary electrode, and saturated Ag/AgCl as reference electrode. Potentials were reported against the reversible hydrogen electrode (RHE) based on the equation: $E_{RHE} = E_{Ag/AgCl} + (0.197 + 0.059 × pH)$ V. Potentials were reported against the normal hydrogen electrode (NHE) based on the equation: $E_{NHE} = E_{Ag/AgCl} + 0.197$ V. In addition, the Ag/AgCl electrode was routinely refilled with fresh saturated KCl electrolyte and calibrated by $[Fe(CN)_6]^{3-}/[Fe(CN)_6]^{4-}$ redox couple for accuracy. Typically, 2 mg catalysts and 20 µL Nafion were added into 1 mL water-ethanol (2: 1) solution. The solution was ultrasonically treated for 30 min. Then, 6 µL of the sample solution was dropped on the effective working area of the glassy carbon electrode. The catalyst loading on the glassy carbon working electrode was 0.17 mg•cm⁻². Cyclic voltammetry (CV) curves were recorded in 0.05 M phosphate buffered saline (PBS) solution (pH = 6.99) at a scan rate of 50 mV•s⁻¹ with iR-compensation. Differential pulse voltammetry (DPV) curves were performed at amplitude of 0.005 V in 0.05 M PBS with iR-compensation. Square wave voltammetry (SWV) curves were carried out at amplitude of 0.025 V in 0.05 M PBS with iR-compensation. The capacitances of the KMnPO₄ and KMnPO₄•H₂O catalysts were obtained by CVs in the non-Faradaic potential region (0.79 V−0.89 V vs RHE). The scan rates were 20, 40, 60, 80, 100, 120, 140, 160, 180, and 200 mV•s⁻¹. The Tafel plots were estimated by linear sweep voltammetry (LSV) measurements at a scan rate of 0.5 mV•s⁻¹ in 0.05 M PBS

with iR-compensation. The differential pulse voltammetry current responses of the catalysts at different pH were recorded in 0.05 M PBS (pH = 6.79, 6.99, 7.21, 7.38, 7.58) at a scan rate of 50 mV•s⁻¹. The square wave voltammetry current responses of the catalysts at different electrolyte concentrations were recorded at a scan rate of 50 mV•s⁻¹. DPV curves were also performed at amplitude of 0.005 V in CH₃CN/ 0.05 M $n$-Bu₄NPF₆ with iR-compensation. Controlled potential electrolysis (CPE) was used to evaluate the stability of the catalyst under the same experimental setup without iR-compensation. The dependence of the OER currents on phosphate concentration were studied in phosphate buffers with different concentrations. Na₂SO₄ was selected as the supplementary electrolyte. We provided the resistance values of the electrolyte solutions involved in electrochemical tests (Supplementary Fig. 30). The error bars were the standard deviations of three repeated measurements.

## Computational studies

We utilized first-principles methodology to conduct the density functional theory (DFT) calculations on the basis of the generalized gradient approximation (GGA) with the Perdew-Burke-Ernzerhof (PBE) formulation[77,78]. To represent the ionic cores, we have selected the projected augmented wave (PAW) potentials, which provide an efficient and accurate means of capturing the electronic structure of the cores[79]. Additionally, we have also considered valence electrons using a plane wave basis set at a kinetic energy cutoff of 520 eV, ensuring that the valence electron dynamics are applicable. To account for partial occupancies of Kohn−Sham orbitals, we applied the Gaussian smearing method with a width of 0.03 eV. Self-consistency was believed when the difference in electronic energy was below $10^{-6}$ eV. For geometry optimization, convergence was believed when the energy change fell below 0.05 eV Å⁻¹. In our structural analysis, we incorporated $U$ correction for Mn atoms at a value of 4.67. Brillouin zone integration was performed using a Monkhorst-Pack k-point sampling scheme with dimensions of $2 × 2 × 1$ for each structure considered in this study. The vacuum spacing perpendicular to the structure's plane was set at 18 Å for surfaces. To calculate free energy ($G$), we employed $G = E_{ads} + ZPE − TS$ equation where $G$ denotes free energy, while ZPE and TS refer to zero point energy and entropic contributions, respectively. Lastly, adsorption energies ($E_{ads}$) were determined using $E_{ads} = E_{ad/sub} − E_{ad} − E_{sub}$, where $E_{ad/sub}$ represents the total energy of an optimized adsorbate/substrate system, while $E_{ad}$ and $E_{sub}$ correspond to the total energies of isolated adsorbates within their respective structures and clean substrates, respectively.

# Data availability

The data that support the findings of this study are available either from the Supplementary Information or from the corresponding author upon reasonable request.

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

## Acknowledgements

We are grateful for the Starting Research Funds of Shaanxi Normal University, the National Natural Science Foundation of China (Grant Nos. 22279159 (Y.Y.) & 22379088 (W.Z.)) and Excellent Graduate Training Program of Shaanxi Normal University (Grant No. LHRCYB23005 (S.Y.)). We gratefully acknowledge the BL14W1 beamline of Shanghai Synchrotron Radiation Facility (SSRF) Shanghai, China, for providing the beam time.

## Author contributions

Experiments were conceived and designed by W. Z. and Y. Y.. S. Y. and W. Z. co-wrote the manuscript, with inputs from all authors. S. Y. performed the electrochemistry and characterization experiments. K. Y. and Y. Y. contributed to the in-situ XAS studies. R. C. and H. Z. analyzed the kinetic results and provided help in the theoretical studies. X. L. and S. L. provided help in physical characterizations.

## Competing interests

The authors declare no competing interests.
