## [Peer Review File · Nature Communications]

REVIEWER COMMENTS

Reviewer #1 (Remarks to the Author):

The authors studied the OER (oxygen evolution reaction) mechanisms of two different Mn-based catalysts with distinct coordination structures. The $\text{KMnPO}_4 \cdot \text{H}_2\text{O}$ that has crystalline water (originally coordinated water) involves a six-coordinated $[\text{MnO}_6]$ motif during the whole catalytic cycles. On the contrary, a five-coordinated $[\text{MnO}_5]$ motif is involved during water oxidation by KMnPO_4 , which exhibits higher intrinsic OER activity. The detailed reaction mechanisms are revealed combining many characterization methods. However, both catalysts showed poor OER activity, compared to the state-of-the-art Mn-based OER catalysts. The obtained mechanistic information might be only specific for the catalysts investigated by the authors, instead of having generality. There are also some problems of the present data (which will be commented as below). Therefore, this reviewer suggests that the manuscript should be published in a more specialized journal, but is not suitable for Nature Communications.

1. The authors said that two catalysts share many physical similarities except their distinct Mn coordination structures. However, there should be also some other parameters, e.g. conductivity, which would significantly determine the OER activity. Low coordinated metal oxide catalyst may have many oxygen vacancies and therefore have higher conductivity (ACS Catal. 2023, 13, 6000). The authors should consider if conductivity is a main factor to determine OER activity in this research work.

2. The OER activity test and the electrochemical kinetic study were performed in 0.05 M PBS (pH 7). However, the concentration of buffer and the ionic strength is low. This will result a significant pH gradient and make the OER current be significantly influenced by mass transport. Moreover, low ionic strength cannot remove the diffuse double layer effects (J. Am. Chem. Soc. 2010, 132, 16501).

3. In neutral OER, buffer electrolyte usually plays important role to promote proton transfer (atom-proton transfer mechanism, see: PNAS 2010, 107, 7225). Therefore, the buffer concentration dependence should be studied, which will also help to propose correct mechanisms. Noted that when studying buffer concentration dependence, excessive inert supporting electrolyte should be added to offer a high ionic strength, when the buffer concentration is low.

4. The authors declared that “the structure flexibility of $[\text{MnO}_5]$ is thermodynamically favored in stabilizing Mn(III)-OH and generating Mn(V)=O ”. According to the experimental data, there are more Mn(IV) and Mn(III) generated for KMnPO_4 catalyst. This result just indicates in KMnPO_4 the manganese is easily oxidized compared to $\text{KMnPO}_4 \cdot \text{H}_2\text{O}$, instead of stabilizing Mn(III)-OH for KMnPO_4 . Actually the oxidation state of Mn is +4 or even higher under the OER condition where the Mn(III)-OH is not stable.

5. In Page 7 of the maintext, the authors said “Although the onset potentials of water oxidation do not change with the pH values of the electrolyte, the catalytic currents are dependent on the pH values.” The “onset potential” is an ambiguous concept, it is only meaningful when the condition is defined. For example, one can define the potential reach to 10 uA/cm^2 or 0.1 mA/cm^2 as the catalytic onset. In this case, the onset potential should also be pH dependent.

6. There are some problems of XAS data interpretation and fitting. The average Mn oxidation state should be calculated using the spectra of the reference Mn oxide compounds, instead of deriving from EXASf fitting. Therefore, the evidence of generation of Mn(V) is not enough. Noted that oxidation of Mn(IV)=O can also generate a Mn(IV)-O radical, instead of Mn(V)=O. The EXAFS data is also questionable, the signal at $\sim 1.2 \text{ \AA}$ might be related to the poor data quality as it is too short for Mn=O bond. The length of Mn=O bond is certainly less than Mn-OH bond, but the length of shrinking is quite high in this study. One previous literature (J. Am. Chem. Soc. 2017, 139, 2277) showed that length of octahedral Mn-O was 1.79 \AA , and the length of Mn=O is 1.70 \AA .

7. Many literatures suggested that the non-oxide OER catalysts are just the precursors and the real catalysts are metal oxide/oxyhydroxide. The authors should provide the content of P before and after reaction to understand whether this material is just a precatalyst or not. Supplementary Fig. 11 indicates that there is an obvious induction period with current increment, which might be related to catalyst reconstruction during OER.

Reviewer #2 (Remarks to the Author):

S. Yang et al. performed a mechanistic study to identify the reaction mechanisms of OER on Mn-based catalyst. They find that Mn(V)=O is responsible for O-O bond formation and its concentration determines the intrinsic activity. Basically, the authors have conducted variable characterizations to support their arguments. However, the proposed mechanism is not sufficient to describe the in-situ OER process. Some conclusions seem to be proposed by speculations. The DFT calculations also show some inconsistencies with the proposed mechanism. A major revision is necessary before it might be considered for publication. Below are some detailed comments:

1. A key point of OER is the in-situ reconstruction of the pre-catalyst, the discussion of which is missing. If surface reconstruction occurs, all the discussions may be unreliable. The authors must provide solid evidence to exclude the influence of reconstruction.
2. The analysis of Figure 6e is incorrect. From Figure 6e, the peaks exhibit an obvious right shift with the increase of voltage, which indicates the shortening of bond length and formation of low valence Mn. The authors should re-analyze the derived information from Figure 6e.
3. How does the potential into RHE/NHE at different pH? Is it by equation or reference electrode calibration? Is it appropriate to use Ag/AgCl at different pH? In general, Ag/AgCl is used in acid and saturated calomel electrode (SCE) is preferred for the test crossing different pH.
4. The conclusion that the Mn(V)=O species is at equilibrium with Mn(IV)=O seem to be speculation. Direct experimental evidence needs to be provided to justify it.
5. Basically, the free energy diagram should illustrate all the charge transfer steps within the reaction. Based on the proposed mechanism in Figure 6h, the first charge transfer step occurs at the water

dissociation step, which is not involved in the free energy diagram in Figure 7b. The free energy diagram should be re-calculated in a consistent manner with the proposed mechanism.

6. The free energy pathway for $\text{KMnPO}_4 \cdot \text{H}_2\text{O}$ should be provided to make a direct comparison with that of KMnPO_4 . For Mn-O species, it is unreasonable to conclude the same Mn valence state under different coordination. I would like to see the Bader charge analysis results of the Mn-O state for both $\text{KMnPO}_4 \cdot \text{H}_2\text{O}$ and KMnPO_4 .

Reviewer #3 (Remarks to the Author):

In this study, the authors synthesized $\text{KMnPO}_4 \cdot \text{H}_2\text{O}$ and its dehydrated counterpart, KMnPO_4 , as an electrocatalytic catalyst in water oxidation. The suggested catalysts, which possess 6- and 4-coordinated Mn centers respectively, were employed to understand the role of Mn coordination in electrocatalytic water oxidation. Through in-depth electrochemical and spectral analyses, the authors investigated the structure-performance correlation and underlying water oxidation mechanisms in Mn-based catalytic system. While their findings are intriguing, the current manuscript raises several questionable uncertainties, and some of the conclusions are not fully supported by acceptable data/analysis and seem to be overinterpreted. Therefore, I cannot recommend the publication of this work at this stage, and the authors should provide further clarifications for the following issues.

1. Since this study aims to reveal structure-performance correlations, it is crucial that more direct analytical evidence regarding the crystal structure of the proposed catalysts is to be provided, e.g., high-resolution TEM images with proper d-spacings indicated, details of simulated patterns from XRD, etc.
2. In Figure 3b, the authors claimed the obvious presence of an Mn^{III/IV} oxidation peak in both catalysts. This observation, however, seems to be contentious, particularly regarding $\text{KMnPO}_4 \cdot \text{H}_2\text{O}$.
3. The authors investigated the Mn(IV) species using EPR measurements and UV-vis analysis. In the UV-vis results, the signal indicative of Mn(IV) prominently increases with rising potential for KMnPO_4 , whereas that of $\text{KMnPO}_4 \cdot \text{H}_2\text{O}$ remains almost unchanged. However, the EPR spectra did not showcase such opposite tendency observed in the UV-vis data. Is there any reason for this phenomenon?
4. In Figures 4e and 4f, the authors argued that new signals appear in the 700 to 800 cm^{-1} region. However, the as-claimed signals seem to be very weak and even appear to be noise-like. Further detailed

analysis of these peaks is necessary. Additionally, the outcomes denoted by these peaks do not seem to (or only partially) match well with those of the EPR and UV-vis analyses.

5. When investigating the oxidation states of metal through XANES spectral analysis, it is necessary to confirm the energy shift with well-known reference compounds of distinct Mn oxidation states.

6. The authors elucidated the OER mechanism by bond formation changes between Mn and oxygen in the respective catalyst based on the Fourier transform spectra at varying potentials. However, the claim for such interpretation seems unsupported and should be further elaborated with proper experimental evidences (e.g., isotopic labeling experiments) and relevant references.

7. In order to compare the catalytic activity and mechanism at the 4- and 6-coordinated Mn centers, the Gibbs free energy changes for $\text{KMnPO}_4 \cdot \text{H}_2\text{O}$ should be presented alongside.

Response to Reviewers

Point-by-point response to referee's comments:

Reviewer #1:

The authors studied the OER (oxygen evolution reaction) mechanisms of two different Mn-based catalysts with distinct coordination structures. The $\text{KMnPO}_4 \cdot \text{H}_2\text{O}$ that has crystalline water (originally coordinated water) involves a six-coordinated $[\text{MnO}_6]$ motif during the whole catalytic cycles. On the contrary, a five-coordinated $[\text{MnO}_5]$ motif is involved during water oxidation by KMnPO_4 , which exhibits higher intrinsic OER activity. The detailed reaction mechanisms are revealed combining many characterization methods. However, both catalysts showed poor OER activity, compared to the state-of-the-art Mn-based OER catalysts. The obtained mechanistic information might be only specific for the catalysts investigated by the authors, instead of having generality. There are also some problems of the present data (which will be commented as below). Therefore, this reviewer suggests that the manuscript should be published in a more specialized journal, but is not suitable for Nature Communications.

We thank the reviewer very much for the valuable comments on our manuscript. The intent of our manuscript is to get a clear understanding of the kinetics of water oxidation, which is an important topic that still needs substantial progress. For kinetic studies on heterogeneous electrocatalysis, a prerequisite is a clear crystal structure of the catalyst. Thus, such studies can only be finely realized based on catalysts with good crystallinity. Under such circumstances, it is understandable that the observed activity from crystals is not as good as those nano-engineered materials. For the mechanism of water oxidation, it is still very ambiguous even in molecular catalysis, let alone heterogeneous electrocatalysis with more uncertainties. At the current stage of this research area, it is still far from getting a mechanism with generality. In contrast, any solid progress in this area is momentous to piecing together the puzzle of water oxidation. We hope the reviewer can agree with us on this point. We addressed all the points raised by the reviewer as summarized below.

1. The authors said that two catalysts share many physical similarities except their distinct Mn coordination structures. However, there should be also some other parameters, e.g. conductivity, which would significantly determine the OER activity. Low coordinated metal oxide catalyst may have many oxygen vacancies and therefore have higher conductivity (ACS Catal. 2023, 13, 6000). The authors should consider if conductivity is a main factor to determine OER activity in this research work.

We thank the reviewer for the helpful and constructive suggestion. We supplemented the conductivity and hydrophilicity analysis of the catalysts in the new Supplementary Fig. 7 and the new Supplementary Fig. 8, respectively. The conductivity and hydrophilicity are similar between the two catalysts, which avoids unfair comparison of their OER activities. Relevant discussions have been added in the revised manuscript on Page 5 Line 10-13, highlighted with a yellow background. The provided valuable reference has been cited as ref. 45.

2. The OER activity test and the electrochemical kinetic study were performed in 0.05 M PBS (pH = 7).

However, the concentration of buffer and the ionic strength is low. This will result a significant pH gradient and make the OER current be significantly influenced by mass transport. Moreover, low ionic strength cannot remove the diffuse double layer effects (J. Am. Chem. Soc. 2010, 132, 16501).

We thank the reviewer for the helpful and constructive suggestion. In this manuscript, the concentrations of conjugated base and conjugated acid are both 0.05 M. The analysis in the manuscript is based on the kinetic currents, which are low and not limited by mass transport. We tested the electrocatalysis in PBS with different concentrations; please kindly see Q3 for details. The provided valuable reference has been cited as ref. 47.

3. In neutral OER, buffer electrolyte usually plays important role to promote proton transfer (atom-proton transfer mechanism, see: PNAS 2010, 107, 7225). Therefore, the buffer concentration dependence should be studied, which will also help to propose correct mechanisms. Noted that when studying buffer concentration dependence, excessive inert supporting electrolyte should be added to offer a high ionic strength, when the buffer concentration is low.

We thank the reviewer for the helpful and constructive suggestion. As shown in the new Supplementary Fig. 9, the concentration dependence of the buffer solution was studied. As suggested, Na₂SO₄ was selected as the supplementary electrolyte. The OER currents remained basically unchanged, elucidating that proton transfer is not the factor to affect the comparison of the catalytic performance. The result is consistent with the studies for CoPi (J. Am. Chem. Soc. 2010, 132, 16501). Relevant discussions have been added in the revised manuscript on Page 5 Line 13-16, highlighted with a yellow background. The provided valuable reference has been cited as ref. 48.

4. The authors declared that “the structure flexibility of [MnO₅] is thermodynamically favored in stabilizing Mn(III)–OH and generating Mn(V)=O”. According to the experimental data, there are more Mn(IV) and Mn(III) generated for KMnPO₄ catalyst. This result just indicates in KMnPO₄ the manganese is easily oxidized compared to KMnPO₄•H₂O, instead of stabilizing Mn(III)–OH for KMnPO₄. Actually the oxidation state of Mn is +4 or even higher under the OER condition where the Mn(III)–OH is not stable.

We thank the reviewer for the helpful and constructive suggestion. According to the DPV results in Fig. 3b, the oxidation peak areas of Mn^{II/III} in the two catalysts are basically equal, indicating that the electrochemical oxidation generated equal amount of Mn(III) in the two catalysts. However, the second precatalytic oxidation (Mn^{III/IV}) events of the two catalysts differ greatly. Also, the UV-vis spectra in Fig. 3c indicated higher concentration of Mn(III) in KMnPO₄ after the first oxidation. These results showed that the Mn(III) species could be efficiently stabilized in KMnPO₄ with a 5-coordination structure. We calculated the average lattice distortion index of Mn atom of [MnO₅] in KMnPO₄ and [MnO₆] in KMnPO₄•H₂O. The index for the former (0.0487) is higher than that of the latter (0.0426). Based on the above results, it can be determined that the structure of [MnO₅] in KMnPO₄ is thermodynamically favored in stabilizing Mn(III)–OH. Relevant discussions have been added into the revised manuscript on Page 12 Line 16-19, highlighted with a yellow background.

5. In Page 7 of the main text, the authors said “Although the onset potentials of water oxidation do not change with the pH values of the electrolyte, the catalytic currents are dependent on the pH values.” The “onset potential” is an ambiguous concept, it is only meaningful when the condition is defined. For example, one can define the potential reach to $10 \mu\text{A}/\text{cm}^2$ or $0.1 \text{ mA}/\text{cm}^2$ as the catalytic onset. In this case, the onset potential should also be pH dependent.

We thank the reviewer for the helpful and constructive suggestion. The Reviewer is right. We cannot accurately judge the specific catalysis initiation potential. At the same time, we observed the pH response of the two catalysts under low current conditions. At currents of 10 and 25 μA , both catalysts have already started to catalyze and the catalytic potentials are linearly related to the pH of the electrolyte solutions (new Supplementary Fig. 14). We modified the descriptions on Page 8 Line 5-9 accordingly, which does not affect the conclusions of the manuscript.

6. There are some problems of XAS data interpretation and fitting. The average Mn oxidation state should be calculated using the spectra of the reference Mn oxide compounds, instead of deriving from EXASf fitting. Therefore, the evidence of generation of Mn(V) is not enough. Noted that oxidation of Mn(IV)=O can also generate a Mn(IV)-O radical, instead of Mn(V)=O. The EXAFS data is also questionable, the signal at $\sim 1.2 \text{ \AA}$ might be related to the poor data quality as it is too short for Mn=O bond. The length of Mn=O bond is certainly less than Mn-OH bond, but the length of shrinking is quite high in this study. One previous literature (J. Am. Chem. Soc. 2017, 139, 2277) showed that length of octahedral Mn-O was 1.79 \AA , and the length of Mn=O is 1.70 \AA .

We thank the reviewer for the helpful and constructive suggestion. Firstly, the Fourier transform (FT) spectra were not phase-corrected, and the bond length would be much shorter than the real value. Secondly, the length of Mn-O in manganese phosphates is longer than that in manganese oxides, as the latter has more ionic bond character. In our system, the oxygen in Mn-O is from phosphate, while the oxygen in Mn=O is an absolute oxygen atom from water. In addition, the Mn=O in the given literature on MnO (J. Am. Chem. Soc. 2017, 139, 2277) is Mn(IV)=O, which is longer than Mn(V)=O. Thus, the bond length of shrinking from Mn-O to Mn=O will be much more significant in our system than that in MnO.

For the Mn(IV)-O' radical query, the oxidation of Mn(IV)=O can indeed generate a Mn(IV)-O' radical. However, the Mn(IV)-O' and Mn(IV)=O are two valence tautomeric forms at equilibrium. It is very challenging to distinguish them even if it is possible. For details, please kindly see our recent works: Chem. Soc. Rev. 2021, 50, 4804, from Rui Cao in collaboration with Prof. Kallol Ray and Prof. Wonwoo Nam; J. Am. Chem. Soc. 2021, 143, 14613, from Wei Zhang and Rui Cao in collaboration with Prof. Shunichi Fukuzumi and Prof. Wonwoo Nam). In the revised manuscript, we described the Mn(V)=O in a formal oxidation state manner, which is acceptable in literature and would not affect the conclusions in our manuscript. Relevant discussions have been added in the revised manuscript on Page 9 Line 12-14, highlighted with a yellow background.

The presence of Mn(V) is determined by electrochemical experiments. As shown in Fig. 5a, three successive oxidation events have been observed, corresponding to $\text{Mn}^{\text{II/III}}$, $\text{Mn}^{\text{III/IV}}$, and $\text{Mn}^{\text{IV/V}}$.

The results of the XAS showed that the energy of the absorption side increased successively, indicating that the oxidation state of Mn increased when the potential increased continuously. For crystals with poor conductivity, the in-situ spectral data contrasts sharply with those ex-situ spectra for high-valent oxides. We supplemented the ex-situ XAS studies of the two manganese phosphates and MnO in the new Supplementary Fig. 20. The valence states and coordination environments have been discussed in the revised manuscript on Page 9 Line 22-33, highlighted with a yellow background.

7. Many literatures suggested that the non-oxide OER catalysts are just the precursors and the real catalysts are metal oxide/oxyhydroxide. The authors should provide the content of P before and after reaction to understand whether this material is just a pre-catalyst or not. Supplementary Fig. 11 indicates that there is an obvious induction period with current increment, which might be related to catalyst reconstruction during OER.

We thank the reviewer for the helpful and constructive suggestion. We performed the Inductive Coupled Plasma (ICP) analysis of catalysts after electrolysis. There are technique issues in the analysis of P element by ICP. Because the in-situ mass of P (mass/charge ratio = 31) is subjected to strong background interference from $^{14}\text{N}^{16}\text{O}^+\text{H}^+$, $^{15}\text{N}^{16}\text{O}^+$, and $^{14}\text{N}^{17}\text{O}^+$. The contents of P element are determined to be even higher than the theoretical values, which can't help us to accurately quantify the content of P. The Raman spectroscopy is more sensitive and accurate, thus we analyzed the Raman spectra of two catalysts electrolyzed at different potentials for 10 minutes. First, the catalyst was electrolyzed at an open circuit potential for 30 minutes to stabilize the surface environment of the catalyst and then electrolyzed at 1.00 V, 1.20 V, 1.40 V, 1.60 V, and 1.80 V for 10 minutes to collect spectral signals. As shown in the new Fig. 4e-4f, the in-situ Raman signals of the manganese phosphate structure does not change obviously with potential and time, which indicates the structural stability of the catalyst structure during catalysis. Relevant discussions have been added into the revised manuscript on Page 7 Line 37-38 and on Page 9 Line 16-19, highlighted with a yellow background.

Reviewer #2:

S. Yang et al. performed a mechanistic study to identify the reaction mechanisms of OER on Mn-based catalyst. They find that Mn(V)=O is responsible for O–O bond formation and its concentration determines the intrinsic activity. Basically, the authors have conducted variable characterizations to support their arguments. However, the proposed mechanism is not sufficient to describe the in-situ OER process. Some conclusions seem to be proposed by speculations. The DFT calculations also show some inconsistencies with the proposed mechanism. A major revision is necessary before it might be considered for publication. Below are some detailed comments:

We thank the reviewer very much for the valuable comments on our manuscript.

1. A key point of OER is the in-situ reconstruction of the pre-catalyst, the discussion of which is missing. If surface reconstruction occurs, all the discussions may be unreliable. The authors must provide solid evidence to exclude the influence of reconstruction.

We thank the reviewer for the helpful and constructive suggestion. We supplemented the Raman analysis of the catalysts before and after electrolysis in the new Supplementary Fig. 18. The Raman analysis confirms that the surface of the two catalysts is unchanged after electrolysis. In addition, the catalysts were electrolyzed for 10 minutes at different potentials. The collected in-situ Raman signals could accurately determine that the catalyst surface was not reconstructed (new Fig. 4e-4f). The XPS spectra of the two catalysts after electrolysis are displayed in Supplementary Fig. 19. The peak positions of the Mn 3s and Mn 2p spectra are unchanged. Relevant discussions have been added into the revised manuscript on Page 7 Line 37-38 and on Page 9 Line 16-19, highlighted with a yellow background.

2. The analysis of Figure 6e is incorrect. From Figure 6e, the peaks exhibit an obvious right shift with the increase of voltage, which indicates the shortening of bond length and formation of low valence Mn. The authors should re-analyze the derived information from Figure 6e.

We thank the reviewer for the helpful and constructive suggestion. The wide signal at $\sim 1.5 \text{ \AA}$ can be attributed to the Mn–O scattering path in the $[\text{MnO}_4]$ group and Mn–O_w (adsorbed water around the site) scattering path at higher radial distance. The right shift of the peak is considered to be from the Mn–O_w scattering path. At higher potentials that water oxidation can be accomplished, Mn–O₂ adducts with weak Mn–O bond can be formed to exhibit a right shift of the signal. Relevant discussions have been added into the revised manuscript on Page 10 Line 1-2 and on Page 11 Line 3-4 & 8-10, highlighted with a yellow background.

3. How does the potential into RHE/NHE at different pH? Is it by equation or reference electrode calibration? Is it appropriate to use Ag/AgCl at different pH? In general, Ag/AgCl is used in acid and saturated calomel electrode (SCE) is preferred for the test crossing different pH.

We thank the reviewer for the helpful and constructive suggestion. The saturated Ag/AgCl was used as the reference electrode in this study. Potentials were reported against the reversible hydrogen electrode (RHE) based on the equation: $E_{\text{RHE}} = E_{\text{Ag/AgCl}} + (0.197 + 0.059 \times \text{pH}) \text{ V}$. Potentials were reported against the normal hydrogen electrode (NHE) based on the equation: $E_{\text{NHE}} = E_{\text{Ag/AgCl}} + 0.197 \text{ V}$. The selection of reference electrode depends on the pH of the electrolyte. Generally, saturated calomel electrode (SCE) is usually used as the reference electrode in acidic solutions; Ag/AgCl is used as the reference electrode in near neutral solutions; Hg/HgO is used as the reference electrode in alkaline solutions. The DPV/SWV curves were recorded in electrolytes with different pH values under near-neutral conditions in the manuscript. In addition, the Ag/AgCl electrode was routinely refilled with fresh saturated KCl electrolyte and calibrated by $[\text{Fe}(\text{CN})_6]^{3-}/[\text{Fe}(\text{CN})_6]^{4-}$ redox couple for accuracy. Relevant information has been added into the revised Methods section on Page 15.

4. The conclusion that the Mn(V)=O species is at equilibrium with Mn(IV)=O seem to be speculation. Direct experimental evidence needs to be provided to justify it.

We thank the reviewer for the helpful and constructive suggestion. We supplemented the in-situ

UV-vis absorption signals during electrolysis at 1.80 V for different times in the new Supplementary Fig. 16. The absorption peak at approximately 400 nm (the characteristic signal of Mn^{IV}) does not change with time during catalysis, which is sufficient to prove that Mn(IV)=O is at steady state and is at equilibrium with Mn(V)=O . Relevant discussions have been added into the revised manuscript on Page 9 Line 7-9, highlighted with a yellow background.

5. Basically, the free energy diagram should illustrate all the charge transfer steps within the reaction. Based on the proposed mechanism in Figure 6h, the first charge transfer step occurs at the water dissociation step, which is not involved in the free energy diagram in Figure 7b. The free energy diagram should be re-calculated in a consistent manner with the proposed mechanism.

We thank the reviewer for the helpful and constructive suggestion. In KMnPO_4 , a water molecule is first adsorbed on the surface Mn site to form $\text{Mn(II)-H}_2\text{O}$ and then dissociated into a proton and Mn(III)-OH . We provide the free energy from $\text{Mn(II)-H}_2\text{O}$ to Mn(III)-OH (OH^*) in the new Fig. 7. Relevant discussions have been added into the revised manuscript on Page 13 Line 20-28, highlighted with a yellow background.

6. The free energy pathway for $\text{KMnPO}_4\cdot\text{H}_2\text{O}$ should be provided to make a direct comparison with that of KMnPO_4 . For Mn-O species, it is unreasonable to conclude the same Mn valence state under different coordination. I would like to see the Bader charge analysis results of the Mn-O state for both $\text{KMnPO}_4\cdot\text{H}_2\text{O}$ and KMnPO_4 .

We thank the reviewer for the helpful and constructive suggestion. We supplemented the free energy pathway for $\text{KMnPO}_4\cdot\text{H}_2\text{O}$ in the new Fig. 7. The formation of $-\text{OOH}^*$ is 1.91 eV for $\text{KMnPO}_4\cdot\text{H}_2\text{O}$, which is higher than that of KMnPO_4 (1.66 eV). The OER is thermodynamically favored in KMnPO_4 , as the free energy changes are more evenly distributed in KMnPO_4 catalyst (ChemCatChem 2011, 3, 1159). We also supplemented the Bader charge analysis in the new Supplementary Table 1 and Table 2. The Bader charges of the Mn-O structure for KMnPO_4 and $\text{KMnPO}_4\cdot\text{H}_2\text{O}$ are -1.452 and -1.488, respectively. The proximity of the Bader charges of the two catalysts indicates that the valence states of Mn sites are similar. Relevant discussions have been added into the revised manuscript on Page 13 Line 1-5 & 20-28, highlighted with a yellow background.

Reviewer #3:

In this study, the authors synthesized $\text{KMnPO}_4\cdot\text{H}_2\text{O}$ and its dehydrated counterpart, KMnPO_4 , as an electrocatalytic catalyst in water oxidation. The suggested catalysts, which possess 6- and 4-coordinated Mn centers respectively, were employed to understand the role of Mn coordination in electrocatalytic water oxidation. Through in-depth electrochemical and spectral analyses, the authors investigated the structure-performance correlation and underlying water oxidation mechanisms in Mn-based catalytic system. While their findings are intriguing, the current manuscript raises several questionable uncertainties, and some of the conclusions are not fully supported by acceptable data/analysis and seem to be overinterpreted. Therefore, I cannot recommend the publication of this work at this stage, and the authors should provide further clarifications for the following issues.

We thank the reviewer very much for the valuable comments on our manuscript.

1. Since this study aims to reveal structure-performance correlations, it is crucial that more direct analytical evidence regarding the crystal structure of the proposed catalysts is to be provided, e.g., high-resolution TEM images with proper d-spacings indicated, details of simulated patterns from XRD, etc.

We thank the reviewer for the helpful and constructive suggestion. We supplemented the HR-TEM analysis of two catalysts in the new Supplementary Fig. 2. The TEM analysis further confirms that the crystal structures were clear. The diffraction peaks in the XRD patterns were assigned in the revised Fig. 1. Relevant discussions have been added into the revised manuscript on Page 3 Line 3-4 & 8-12, highlighted with a yellow background.

2. In Figure 3b, the authors claimed the obvious presence of an Mn(III/IV) oxidation peak in both catalysts. This observation, however, seems to be contentious, particularly regarding $\text{KMnPO}_4 \cdot \text{H}_2\text{O}$.

We thank the reviewer for the helpful and constructive suggestion. We adopt a segmented testing method (with a potential range of 0.80 V to 1.20 V) to directly monitor the electrochemical signals of Mn(III) to Mn(IV). The occurrence of Mn(III) to Mn(IV) oxidation process can be observed in Supplementary Fig. 12-13.

3. The authors investigated the Mn(IV) species using EPR measurements and UV-vis analysis. In the UV-vis results, the signal indicative of Mn(IV) prominently increases with rising potential for KMnPO_4 , whereas that of $\text{KMnPO}_4 \cdot \text{H}_2\text{O}$ remains almost unchanged. However, the EPR spectra did not showcase such opposite tendency observed in the UV-vis data. Is there any reason for this phenomenon?

We thank the reviewer for the helpful and constructive suggestion. The difference is mainly due to the low content of Mn(IV) in the $\text{KMnPO}_4 \cdot \text{H}_2\text{O}$ catalyst, and the inability of UV-vis spectroscopy to accurately capture the weak signals of Mn(IV) species. However, EPR spectroscopy is much more sensitive and can detect extremely low concentrations of Mn(IV) species. Relevant discussions have been added into the revised manuscript on Page 7 Line 12-13, highlighted with a yellow background.

4. In Figures 4e and 4f, the authors argued that new signals appear in the 700 to 800 cm^{-1} regions. However, the as-claimed signals seem to be very weak and even appear to be noise-like. Further detailed analysis of these peaks is necessary. Additionally, the outcomes denoted by these peaks do not seem to (or only partially) match well with those of the EPR and UV-vis analyses.

We thank the reviewer for the helpful and constructive suggestion. We re-detected the Raman signals of the two catalysts on an in-situ Raman apparatus with better configuration. The new spectral data have been added to Fig. 4e-4f. By changing the laser intensity, spectral collection time, and spot focusing degree, the scattering signal is now more obvious. In both catalysts, the signal of Mn(IV)=O species increased with the gradual increase of applied potential, which is

consistent with the results of EPR and UV-vis. Relevant information have been added into the revised Methods section on Page 15.

5. When investigating the oxidation states of metal through XANES spectral analysis, it is necessary to confirm the energy shift with well-known reference compounds of distinct Mn oxidation states.

We thank the reviewer for the helpful and constructive suggestion. The results of the XAS showed that the energy of the absorption side increased successively, indicating that the oxidation state of Mn increased when the potential increased continuously. For crystals with poor conductivity, the in-situ spectral data contrasts sharply with those ex-situ spectra for high-valent oxides. We supplemented the ex-situ XAS studies of the two manganese phosphates and MnO in the new Supplementary Fig. 20. The valence states and coordination environments have been discussed in the revised manuscript on Page 9 Line 22-33, highlighted with a yellow background.

6. The authors elucidated the OER mechanism by bond formation changes between Mn and oxygen in the respective catalyst based on the Fourier transform spectra at varying potentials. However, the claim for such interpretation seems unsupported and should be further elaborated with proper experimental evidences (e.g., isotopic labeling experiments) and relevant references.

We thank the reviewer for the helpful and constructive suggestion. To further elucidate the oxygen evolution mechanism that occurs on KMnPO_4 and $\text{KMnPO}_4 \cdot \text{H}_2\text{O}$, isotope-labeled operando DEMS measurements were carried out in electrolytes with H_2^{18}O and H_2^{16}O . First, KMnPO_4 and $\text{KMnPO}_4 \cdot \text{H}_2\text{O}$ were loaded on the porous Au film and labeled with ^{18}O by performing three LSV cycles (0.90 to 2.10 V versus RHE) in 0.05 M PBS H_2^{18}O electrolyte. Then, the ^{18}O -labeled catalysts were washed with a large amount of water to remove the adsorbed H_2^{18}O , followed by three cycles in 0.05 M PBS H_2^{16}O electrolyte. The experimental results are shown in the new Fig. 6h-6i and the new Supplementary Fig. 25. The signals detected by the two catalysts in 0.05 M PBS H_2^{18}O electrolyte are mainly the mass spectrum current generated by $^{36}\text{O}_2$. The results show that the conventional adsorbate evolution mechanism (AEM) is dominant in the two catalysts. At the same time, a weak $^{34}\text{O}_2$ current signal was also detected, in which the content ratio of $^{34}\text{O}_2$ to $^{36}\text{O}_2$ in KMnPO_4 catalyst was 5.1%, while the content ratio of $^{34}\text{O}_2$ to $^{36}\text{O}_2$ in $\text{KMnPO}_4 \cdot \text{H}_2\text{O}$ catalyst was 9.5%, further indicating that the coordination water of H_2^{16}O in the 6-coordinate structure participated in the water oxidation process. All the signals obtained by LSV cycling in 0.05 M PBS H_2^{16}O electrolyte after labeling were $^{32}\text{O}_2$ signals, and no $^{36}\text{O}_2$ signals were observed. Relevant discussions and references have been added into the revised manuscript on Page 11 (bottom 5 lines) & Page 12 (top 9 lines), highlighted with a yellow background.

7. In order to compare the catalytic activity and mechanism at the 4- and 6-coordinated Mn centers, the Gibbs free energy changes for $\text{KMnPO}_4 \cdot \text{H}_2\text{O}$ should be presented alongside.

We thank the reviewer for the helpful and constructive suggestion. We supplemented the free energy pathway for $\text{KMnPO}_4 \cdot \text{H}_2\text{O}$ in the new Fig. 7. The formation of $-\text{OOH}^*$ is 1.91 eV for $\text{KMnPO}_4 \cdot \text{H}_2\text{O}$, which is higher than that of KMnPO_4 (1.66 eV). The OER is thermodynamically favored in KMnPO_4 , as the free energy changes are more evenly distributed in KMnPO_4 catalyst

(ChemCatChem 2011, 3, 1159). Relevant discussions have been added into the revised manuscript on Page 13 Line 1-5 & 20-28, highlighted with a yellow background.

Again, we thank the hard work from all the reviewers and their insightful comments that have helped to improve this manuscript significantly.

REVIEWER COMMENTS

Reviewer #1 (Remarks to the Author):

The authors have substantially revised their manuscript and demonstrated the importance of their work. However, some data were still not clearly explained, therefore further explanations/revisions are required.

1. Page 5: "The OER currents remained basically unchanged, elucidating that proton transfer is not the factor to affect the comparison of the catalytic performance". This statement is ambiguous and confusing. The OER performance, not exhibiting significant dependence on buffer concentration, actually suggests that the buffer electrolyte does not promote proton transfer.
2. How did the authors conclude that "the oxidation peak areas of Mn(II/III) in the two catalysts are basically equal" (Page 6)? According to Fig. 3a and 3b (as well as ECSA normalized CV, Fig. S5d, SI), KMnPO_4 clearly shows higher Mn(II/III) peak at ~ 1.25 V vs. RHE, compared to $\text{KMnPO}_4 \cdot \text{H}_2\text{O}$.
3. The authors emphasized again that KMnPO_4 can stabilize Mn(III)-OH. This declaration is not accurate as the term 'stabilize' usually implies resistance to change under the external driving force. However, both KMnPO_4 and $\text{KMnPO}_4 \cdot \text{H}_2\text{O}$ are gradually oxidized when increasing the applied potential. Moreover, the spectroscopic data indicate more high-valent species are generated for KMnPO_4 . Higher concentration of Mn(III) is indeed generated in KMnPO_4 after the first oxidation peak, nevertheless, this 'stabilization' is only effective in a narrow potential window.
4. The authors proposed that water attack to Mn(V)=O to generate Mn(III)-OOH is the rate-determining step (RDS), and Mn(V)=O and Mn(IV)=O are in equilibrium before RDS. According to classical adsorption equilibrium model, such mechanism should have a Tafel slope close to 40 mV/dec (Sci. Rep. 2015, 5, 13801). However, the Tafel slope of both catalysts are nearly 260 mV/dec. Despite the Tafel slope values can be influenced by many factors and be deviated from theoretical values, the authors should provide some reasonable explanations for these unusually high Tafel slope values.
5. The data presented in Fig. S10, SI requires clarification. Why the areas of the mixed Mn(II/III) and Mn(III/IV) reduction peaks are decreased when the initial potential of scan goes to more positive values?
6. An explanation is needed for the observed variation in pH dependence at different current densities for $\text{KMnPO}_4 \cdot \text{H}_2\text{O}$ (Fig. S14b and Fig. S15b).
7. Some typos like "Mn(VI)=O" in Page 11 should be corrected.

Reviewer #2 (Remarks to the Author):

Some of the issues have been solved in the revised version. However, there are still some remaining concerns that need to be properly addressed before it can be recommended for publication.

1. From the results in Fig. 6a, the valence state of Mn is indeed increasing. But it remains highly questionable to attribute the enlargement of bond length to the weakening of OO adsorption. Basically, XAS reflects the information of bulk materials rather than surface adsorption. Besides, it is unreasonable to judge the change of the characteristic peak to one single adsorbate since multiple adsorption species exist simultaneously at the surface domain. The authors should re-analyze the information from XAS.

2. The authors say that they provide the free energy from Mn(II)–H₂O to Mn(III)–OH (OH*) in the new Fig. 7. However, I cannot see any change of the free energy diagram between the old Fig. 7 and the new Fig. 7. The authors should explain this.

Reviewer #3 (Remarks to the Author):

The authors properly addressed most of the comments and concerns raised by the reviewer. The manuscript is now recommended for publication in Nature communications.

Response to Reviewers

Point-by-point response to referee's comments:

Reviewer #1:

The authors have substantially revised their manuscript and demonstrated the importance of their work. However, some data were still not clearly explained, therefore further explanations/revisions are required.

We thank the reviewer very much for the valuable comments on our manuscript. We addressed all the points raised by the reviewer as summarized below.

1. Page 5: “The OER currents remained basically unchanged, elucidating that proton transfer is not the factor to affect the comparison of the catalytic performance”. This statement is ambiguous and confusing. The OER performance, not exhibiting significant dependence on buffer concentration, actually suggests that the buffer electrolyte does not promote proton transfer.

We thank the reviewer for the helpful and constructive suggestion. Accordingly, we modified the statement in the manuscript to the following content on Page 5 Line 14-16: “The OER currents remained basically unchanged, elucidating that proton transfer and the OER rate are not affected by the buffer concentration.”.

2. How did the authors conclude that “the oxidation peak areas of Mn(II/III) in the two catalysts are basically equal” (Page 6)? According to Fig. 3a and 3b (as well as ECSA normalized CV, Fig. S5d, SI), KMnPO_4 clearly shows higher Mn(II/III) peak at ~ 1.25 V vs. RHE, compared to $\text{KMnPO}_4 \cdot \text{H}_2\text{O}$.

We thank the reviewer for the helpful and constructive suggestion. The misunderstanding was caused by the background non-Faradaic currents. Compared with CV, DPV can reduce background interference. We performed area integration of the oxidation peaks in Fig. 3b (DPV) to demonstrate their close peak areas. Relevant discussions have been added into the revised manuscript on Page 5 Line 19-20: “Compared with CV, DPV can reduce background interference. The area integration of the oxidation peaks in Fig. 3b demonstrates their close peak areas.”.

3. The authors emphasized again that KMnPO_4 can stabilize Mn(III)–OH. This declaration is not accurate as the term ‘stabilize’ usually implies resistance to change under the external driving force. However, both KMnPO_4 and $\text{KMnPO}_4 \cdot \text{H}_2\text{O}$ are gradually oxidized when increasing the applied potential. Moreover, the spectroscopic data indicate more high-valent species are generated for KMnPO_4 . Higher concentration of Mn(III) is indeed generated in KMnPO_4 after the first oxidation peak, nevertheless, this ‘stabilization’ is only effective in a narrow potential window.

We thank the reviewer for the helpful and constructive suggestion. We used the term “stabilize” to describe the stabilization of Mn(III) against the Jahn-Teller distortion-caused disproportionation of Mn(III). To be more precise, we modified the descriptions accordingly in

manuscript, such as “has more remained Mn(III) species against the Jahn-Teller distortion-caused disproportionation”; “the Mn(III) species can be effectively retained in KMnPO_4 ”; “favored in retaining the generated Mn(III)–OH”.

4. The authors proposed that water attack to Mn(V)=O to generate Mn(III)–OOH is the rate-determining step (RDS), and Mn(V)=O and Mn(IV)=O are in equilibrium before RDS. According to classical adsorption equilibrium model, such mechanism should have a Tafel slope close to 40 mV/dec (Sci. Rep. 2015, 5, 13801). However, the Tafel slope of both catalysts are nearly 260 mV/dec. Despite the Tafel slope values can be influenced by many factors and be deviated from theoretical values, the authors should provide some reasonable explanations for these unusually high Tafel slope values.

We thank the reviewer for the helpful and constructive suggestion. The provided valuable reference shows very detailed kinetic analysis of Tafel slope. Indeed, the Tafel slope is very complicate and can be affected by many factors in OER. The only way to get a reliable Tafel slope of OER is to obtain the detailed mechanism of OER and the dependence of the surface intermediate concentration on potential. In our case, both catalysts showed Tafel slopes of nearly 260 mV/dec. The reasons might be that (1) the oxidation currents of Mn(IV) to Mn(V) interfered the OER currents and (2) the formation of Mn–OOH is a pure chemical step instead of an electrochemical step. When the rds of an electrochemical process is a pure chemical step, the Tafel slope will be very high theoretically. Relevant discussions have been added into the revised manuscript on Page 8 Line 16-18.

5. The data presented in Fig. S10, SI requires clarification. Why the areas of the mixed Mn(II/III) and Mn(III/IV) reduction peaks are decreased when the initial potential of scan goes to more positive values?

We thank the reviewer for the helpful and constructive suggestion. The phenomenon is because that the formation of Mn(IV) or oxygen evolution happens at high potentials. The generation of Mn(IV) and oxygen consumes Mn(III) species. The Mn^{III/IV} redox is quasi-reversible and water oxidation is irreversible in our case. Thus, when the electrode was set at higher potentials, the Mn(III) concentration would be lower to show a decreased cathodic peak of Mn(III) to Mn(II). Relevant discussions have been added into the revised manuscript on Page 7 Line 7-11.

6. An explanation is needed for the observed variation in pH dependence at different current densities for $\text{KMnPO}_4 \cdot \text{H}_2\text{O}$ (Fig. S14b and Fig. S15b).

We thank the reviewer for the helpful and constructive suggestion. This phenomenon is because that these kinetic currents are probably overlapped with the Mn(IV) to Mn(V) oxidation currents, which are independent of pH. At different regions, the pH dependences of catalytic currents, therefore, are slightly different. Relevant discussions have been added into the revised manuscript on Page 8 Line 13-15.

7. Some typos like “Mn(VI)=O” in Page 11 should be corrected.

We thank the reviewer for the helpful and constructive suggestion. We have corrected “Mn(VI)=O” to “Mn(IV)=O” in the revised manuscript.

Reviewer #2:

Some of the issues have been solved in the revised version. However, there are still some remaining concerns that need to be properly addressed before it can be recommended for publication.

We thank the reviewer very much for the valuable comments on our manuscript. We addressed all the points raised by the reviewer as summarized below.

1. From the results in Fig. 6a, the valence state of Mn is indeed increasing. But it remains highly questionable to attribute the enlargement of bond length to the weakening of OO adsorption. Basically, XAS reflects the information of bulk materials rather than surface adsorption. Besides, it is unreasonable to judge the change of the characteristic peak to one single adsorbate since multiple adsorption species exist simultaneously at the surface domain. The authors should re-analyze the information from XAS.

We thank the reviewer for the helpful and constructive suggestion. The *in-situ* XAS was taken under Total Fluorescence Yield (TFY) mode. The XAS results reflect both the information of the bulk material and surface adsorptions. In our case, we tried to get the information of surface adsorptions on the basis of the variations of peaks. At 1.80 V, the O₂ is formed and released. A variety of adsorbents (substrates, intermediates and products) exist in equilibrium with the continuous release of O₂. The re-adsorption of substrate water molecules also happens for any refreshed catalytic site. These adduct together lead to the right shift of the observed signal peak. This right shifted peak centered at the position close to the right shoulder of the broad peak from the sample at 1.00 V, which is consistent with that the catalysis of O₂ formation is completed and part of the catalytic Mn centers go back to its original low valence state. Relevant discussions have been added into the revised manuscript on Page 11 Line 18-23. We also revised relevant descriptions to be more reasonable, such as “the adsorbed water molecules have bonded to the active site (or other species with weak Mn–O bond)”; “the Mn–O_w scattering path (or from species with weak Mn–O bond)”.

2. The authors say that they provide the free energy from Mn(II)–H₂O to Mn(III)–OH (OH*) in the new Fig. 7. However, I cannot see any change of the free energy diagram between the old Fig. 7 and the new Fig. 7. The authors should explain this.

We thank the reviewer for the helpful and constructive suggestion. We are very sorry that we did not clearly express the information in the revised manuscript. We chose the [MnO₄] surface and a separate H₂O as the computational model for KMnPO₄. A water molecule is first adsorbed on the surface Mn site to form Mn(II)–H₂O*, and then dissociates into a proton and Mn(III)–OH*. The Gibbs free energy at this step corresponds to the * + H₂O to OH* free energy change, which is consistent with the first step of the proposed mechanism in Fig. 6j. We modified the descriptions accordingly in the new Fig. 7a and 7c.

Reviewer #3:

The authors properly addressed most of the comments and concerns raised by the reviewer. The manuscript is now recommended for publication in Nature communications.

We thank the reviewer very much.

Again, we thank the hard work from all the reviewers and their insightful comments that have helped to improve this manuscript significantly.

REVIEWERS' COMMENTS

Reviewer #1 (Remarks to the Author):

The authors have properly addressed all of my comments. The manuscript is now recommended for publication.

Reviewer #2 (Remarks to the Author):

The authors have addressed my concerns in the revised version. It can be recommended for publication.